# New 30 m resolution Hong Kong climate, vegetation, and topography rasters indicate greater spatial variation than global grids within an urban mosaic

Brett Morgan[1] and Benoit Guénard[1]

[1]School of Biological Sciences, The University of Hong Kong, Hong Kong SAR

**Correspondence:** Brett Morgan (brettmorgan2@gmail.com)

**Abstract.** The recent proliferation of high quality global gridded GIS datasets has spurred a renaissance of studies in many fields, including biogeography. However, these data, often 1 km at the finest scale available, are too coarse for applications such as precise designation of conservation priority areas and regional species distribution modeling, or purposes outside of biology such as city planning and precision agriculture. Further, these global datasets likely underestimate local climate variations because they do not incorporate locally relevant variables. Here we describe a comprehensive set of 30 m resolution rasters for Hong Kong, a small tropical territory with highly variable terrain where intense anthropogenic disturbance meets a robust protected area system. The data include topographic variables, Normalized Difference Vegetation Index, and interpolated climate variables based on weather station observations. We present validation statistics that convey each climate variable's reliability, and compare our results to a widely used global dataset, finding that our models consistently reflect greater climatic variation. To our knowledge, this is the first set of published environmental rasters specific to Hong Kong. We hope this diverse suite of geographic data will facilitate future environmental and ecological studies in this region of the world, where a spatial understanding of rapid urbanization, introduced species pressure, and conservation efforts is critical. The dataset (Morgan and Guénard, 2018) is accessible at https://doi.org/10.6084/m9.figshare.6791276.

## 1 Introduction

Scale of analysis has long been considered a key concern in biogeographic research (Levin, 1992). Multiple types of scale are relevant to environmental data, including analysis grain, response grain, spatial structure, and study extent (Mertes and Jetz, 2018). Analysis grain, the minimum unit of spatial resolution in a spatial grid, is commonly referred to as a pixel or cell. In research that uses environmental raster data, the pixel size directly dictates the types of biogeographic questions that can be reasonably addressed.

This relationship between analysis grain and study suitability is complex, and higher resolutions are not always advantageous. For example, in global analyses excessively high resolution data would be computationally cumbersome and unnecessary if the goal is to characterize broad patterns. However as shown below, many studies have found notable benefits of higher resolution climatic predictors. Unfortunately, regional analyses lacking local data are limited to using global datasets and the grain size at which they are available (e.g. Cheng and Bonebrake, 2017).

Species distribution modeling (SDM) is a common application of gridded environmental data, where the selected analysis grain has important consequences. In SDM, one or more geographic predictors are associated statistically with the location of known observations of a species (Peterson et al., 2011). The resulting statistical model can be converted to a geographic model: a spatially continuous measure of species occurrence likelihood across the landscape of interest. SDMs are used for many applications, including predicting potential ranges of invasive species, characterizing ecological constraints on species ranges, discovering biodiversity, and planning protected areas (Peterson et al., 2011). The effects of SDM grain size manipulation is an active area of research. Below, we summarize findings on four main effects: estimated distribution size, inclusion of fine scale features, predictor variable selection, and model predictive ability.

Coarser environmental data consistently result in SDMs that predict larger areas of species presence (Connor et al., 2017; Franklin et al., 2013; Seo et al., 2009). Overestimation of SDMs is especially a concern for conservation purposes, where inferred size of suitable habitat is often used to inform extinction risk assessments. Mistakenly large calculated distributions could result in species that are assigned artificially low risk levels.

Coarse resolution predictors can cause SDMs to omit small, but important areas. Particularly of interest are microrefugia, climatically unique patches of land that can harbor rare species, and are especially important for conservation as species distributions respond to climate change (Dobrowski, 2010). Meineri and Hylander (2017) demonstrated that because high resolution climate models included such microrefugia, the resulting species distribution models predicted lower extinction rates for plant species than coarser predictors. Nezer et al. (2016) found that 10 m or 100 m resolution SDMs can reveal other distribution features invisible at lower resolutions (1 km): movement corridors, isolated habitat patches, geomorphologic features, and anthropogenic effects on distributions.

SDM scale can also affect which predictors are selected for model calculation. Certain predictors may be excluded in SDMs because they lack explanatory power at the chosen scale of analysis (Mertes and Jetz, 2017). For example, vegetation measures like the Normalized Difference Vegetation Index (NDVI) in fragmented forests are unlikely to be relevant if the grain size is much larger than the forest patch size, because each grid cell will be a single averaged value. This means that coarse models might not only mischaracterize the distribution pattern itself, but they also may fail to explicate important environmental relationships that determine species occurrence. Indeed, Nezer et al. (2016) found that the most important predictors (vegetation, slope) in their highest resolution models (10 m) were "nearly meaningless" at 1 km resolution. Another study found similar differences in predictor importance related to variation in scale (Lasseur et al., 2006). Of course, predictor importance is always relative and thus is subject to which predictors are included in model building. Therefore this pattern is not expected to be observed in all studies, but should not be overlooked as a potential source of bias.

Last, any consistent effects of SDM grain size on the overall predictive ability of SDMs are unclear. The most commonly used measure of SDM performance is Area Under Curve (AUC), where a higher value indicates a greater ability to differentiate between area the species is present or absent. Some studies found increased SDM resolution resulted in increased AUC (Seo et al., 2009; Nezer et al., 2016), while others found no effect (Pradervand et al. 2014) or mixed effects depending on dataset (Guisan et al., 2007). These studies used different species, predictors, scales, regions, and modeling algorithms, so further research is required to investigate any association between SDM grain size and AUC.

The above advantages of higher resolution environmental data in SDM may be dependent on project-specific factors, such as the quality of species records available and the goals of the research. For example, using environmental grids of a smaller grain size than the locational accuracy of the available species records is untenable. Additionally, stationary species (e.g. lichens) may be more strongly affected by local factors while highly mobile species (e.g. birds) may only be limited at broader scales. Indeed, it has been shown that plant (rather than bird or mammal) species models with highest locational accuracy were those most improved by higher resolution (Guisan et al., 2007). Lastly, the utility of fine grain environmental grids can depend on habitat; flat deserts may have less biologically relevant fine-scale spatial variation compared to mountainous forests or tropical areas fragmented by human activity, like Hong Kong.

In this study, a new series of rasters for Hong Kong are introduced particularly suited for SDM. The layers produced focus on long term climate averages, topography, and vegetation. We asked how the new 30 m scale rasters provide new information on climatic variables in Hong Kong in comparison to a global dataset already available. We hypothesize that our new climate data will indicate greater variation (measured as raster standard deviation) in climate variables. The development of high-resolution environmental rasters is particularly important in tropical regions where species exhibit small distribution ranges (as predicted by Rapoport's Rule: Stevens, 1989) and where understanding interactions between organisms and their changing habitats is paramount.

## 2 Study area: Hong Kong

Geographic data of appropriate resolution is critically important for conducting research within the Hong Kong Special Administrative Region of China, because of its complex landscape. Hong Kong exhibits dramatically variable topography, fitting numerous small islands, dozens of mountain peaks over 500 m, 733 km of coastline, and a human population of over 7 million into a land area of only 1,104 km² (Fig. 1). Seasonally variable monsoon winds deliver equatorial heat and torrential precipitation in summer, while northerly winds carry chilly dry air from continental Asia during the winter (Dudgeon and Corlett, 1994). However, daily temperature fluctuations are attenuated by the surrounding South China Sea and Pearl River Estuary. Hong Kong's terrain typically exhibits a stark bifurcation between some of the most densely constructed areas in the world (Lau and Zhang, 2015) and steep, vegetated slopes. Uninhabited expanses are protected as part of 24 country parks and additional special areas that cover over 40% of the territory's land (Agriculture, Fisheries and Conservation Department, 2017). Even within these more natural areas, a strong disturbance gradient encompasses grasslands, shrublands, evergreen secondary forests, and old-growth *feng shui* woods that have been protected from deforestation. Historically Hong Kong has been largely stripped of its trees, and only since the end of World War II and later the establishment of the Country Park system have large swathes of forest begun to regenerate (Zhuang and Corlett, 1997). However this process is frequently reset by human-induced hill fires, which maintain predominantly upland areas as shrubland or grassland (Marafa and Chau, 1999). Hong Kong harbors several unique and restricted habitats, including mangroves in coastal areas and freshwater wetlands in the far northwest.

Hong Kong climate data is available within a variety of global gridded climate datasets (WorldClim 2 - Fick and Hijmans, 2017; MerraClim - Vega et al., 2017; CHELSA - Karger et al, 2017), but none of these have a resolution higher than 1

km. We suspect those global climate models underestimate variation in local climate values, even after consideration of the coarser scale. Local studies of Hong Kong meteorology have largely focused on characterizing and mitigating the effects of urbanization (e.g. Shi et al., 2018; Wang et al., 2017; Nichol et al., 2014; Liu and Zhang, 2011; Ng, 2009; Giridharan et al., 2004). Unfortunately, it appears the climate of Hong Kong's landscape as a whole has been given little notice, and we are unaware of long-term averaged climate rasters available for the region. Relevant studies that do exist include limited variables, and the data appear to be publicly unavailable. We are additionally unaware of Hong Kong data publicly available for vegetation indices such as NDVI, or topographic data other than elevation. Therefore Hong Kong is in dire need of a comprehensive suite of accessible environmental GIS data, at a resolution finer than 1 km, suitable for species distribution modeling and other local applications. To this end, we developed new, 30 m resolution rasters of topography, NDVI, and 10 interpolated climate variables for each month of the year.

## 3 Methods

All data manipulation and geographic analyses were conducted in the R statistical computing environment (v3.3.2, R Core Team, 2016) using RStudio (v1.0.136, RStudio Team, 2015) unless otherwise noted. Analyses are divided into three broad categories of data products, detailed in the sections below: topographic variables, climate variables, and remote sensing variables. The variables developed were selected based on their utility in environmental research, especially SDM, as well as the availability of appropriate source data. An overview schematic of the data workflow is available in Figure S1.

### 3.1 Topographic variables

Data on the physical characteristics of Hong Kong's landmass were assembled from remote sensing inputs, crowdsourced coastline polygons, and a digital terrain model. The topographic variables developed are coastline, elevation, slope, aspect, terrain roughness, relative elevation, distance to coast, water proximity, and urbanicity.

#### 3.1.1 Coastline

As reclamation of land from the ocean in Hong Kong is ongoing, obtaining current data for the coastline can be challenging. Natural coastline and reservoir vectors were downloaded from OpenStreetMap (2018) and merged in QGIS (v3.01, QGIS Development Team, 2018) to produce a shapefile of polygons representing Hong Kong land area as of January 2018. All output rasters were masked to this area.

#### 3.1.2 Elevation, slope, aspect, and roughness

A 5 m resolution Hong Kong digital terrain model (Lands Department, 2017) was upscaled using bilinear resampling. The resulting 30 m DEM was used as the elevation data throughout the study. Four other topographic predictor layers were derived directly from this DEM: aspect, slope, aspect*slope, and a roughness index. These were calculated using the Hong Kong

elevation raster with the terrain() function in the *raster* R package (Hijmans, 2019), using all 8 neighboring cells (queen case). Aspect was transformed from degrees to a measure of north-south exposure ("northness") by cos(aspect*pi/180).

### 3.1.3 Relative elevation

Relative elevation is a measure of the difference in elevation between the pixel of interest, and the lowest pixel within a given radius. A pixel on a mountain peak has a high relative elevation, while a pixel on a flat plain has a relative elevation of 0 (regardless of its elevation above or below sea level). A set of relative elevation layers for Hong Kong were calculated at multiple scales, following the moving window approach of Bennie et al. (2010). The radii used were 60 m, 120 m, 240 m, 480 m, and 960 m. These layers are expected to be most applicable as measures of surface water drainage, and therefore soil moisture as well. Relative elevation has been used as a covariate in climate interpolation as a proxy for cool air draining (Bennie et al., 2010; Ashcroft and Gollan, 2012), but was not included here as a predictor as Hong Kong lacks large valleys and other sheltered areas where this effect would be most relevant.

### 3.1.4 Distance to coast and water proximity

Water bodies adjacent to land areas can act as temperature buffers, contribute to evaporative cooling (Lookingbill and Urban, 2003), and influence precipitation patterns (Heiblum et al., 2011; Paiva et al., 2011); therefore considering their presence is important for climatic predictions. Here, two different methods were used to quantify water body distribution in Hong Kong: distance to coast and water proximity. A distance to coast raster, measured in meters, was produced using the distance() function in the *raster* R package (Hijmans, 2019) with the Hong Kong coastline shapefile described in section 3.1.1. Distance to coast did not incorporate inland water bodies. Second, water proximity (including inland water bodies) was calculated as the percent of the the area surrounding a given pixel covered by land. A value of 1 means that the area within a given radius is entirely terrestrial, while 0 indicates it is entirely aquatic. Multiple water proximity rasters were calculated with varying radii using a circular moving window approach like that described by Aalto et al. (2017), to represent buffering processes at different scales. The radii used were 0.75 km, 1.5 km, 3 km, 6 km, and 12 km.

### 3.1.5 Urbanicity

Urbanicity rasters were developed because in densely constructed areas, urban heat island effects are expected to influence temperatures (Nichol et al., 2013; Shi et al., 2018), and therefore urbanicity may be an important predictor in climate interpolation. High rise buildings can influence temperature by blocking wind, creating shade, acting as heat sinks, and producing thermal pollution. These effects are particularly relevant for this study, as some of Hong Kong's weather observation stations are adjacent to or inside urban centers. To quantify the distribution of developed area, we used a 30 m resolution dataset of percent impervious surface (Brown de Colstoun et al., 2017), which we expect to strongly correlate with urban development. For use in climate predictions this data was smoothed using a Gaussian moving window, because bulk air temperature is not expected to vary at a granular (30 m) scale. at three buffer scales (sigma = 10, 50, 100), using the focalWeight() and focal()

functions in the *raster* R package (Hijmans, 2019), where type = 'Gauss'. The resulting 'urbanicity' layers were later used as climate predictors. In these rasters, completely impervious locations have a value of 100, while vegetated areas have a value of 0.

## 3.2 Climate variables

Climate interpolators are often faced with the challenge of estimating climate parameters over a large area using sparse weather station observations, at least in part of the region considered (e.g. Hu et al., 2016). In contrast, interpolation in Hong Kong is benefitted by a relatively small geographic area and a quite dense network of weather data provided by dozens of permanent weather stations (Hong Kong Observatory, 2018; see Figure S2). Here we use multiple linear regression to predict geographic climate patterns using weather station training points and raster covariates. This is followed by thin plate spline (TPS) interpo-

lation (see Wahba, 1979) of the regression model residuals. TPS is a widely used approach in climate interpolation (e.g. New et al., 2002; Fick and Hijmans, 2017), which fits a curved surface to irregularly distributed points. This two-step interpolation (regression followed by TPS) was based on the approach of Meineri and Hylander (2017).

Weather station observation data and geographic coordinates were downloaded from the web portal of the Hong Kong Observatory (2018). As the goal was to produce a representation of long-term but modern climate, measurements over 20

15 years (1998 to 2017) were included. To ensure averages were reliable, weather stations were only included for interpolation of each variable if at least 8 years of complete data were available within the 20 year window. The minimum number of stations used for each model is provided in Table 2. Monthly observations of ten variables were obtained: maximum temperature, mean daily maximum temperature, mean daily temperature, mean daily minimum temperature, minimum temperature, mean dew point, mean relative humidity, mean wind speed, mean air pressure, and total rainfall.

Climate interpolation consisted of two main steps. First, a linear model was built for each climate variable for each month of the year. Independent variables were selected by searching the literature for similar studies, and choosing predictors we expected to have an influence on climate at this regional scale. When necessary, each predictor was statistically transformed to approach a normal distribution. The six topographic predictors used as model building candidates were: elevation, log-transformed distance to coast, exponentially transformed fine and coarse water proximity, log-transformed urbanicity (sigma =

50), and 'northness' - the cross product of aspect and slope. The water proximity layers were products of additively combining multiple scale rasters into fewer predictors: fine water proximity was the sum of 0.75 km, 1.5 km, 3 km scale rasters, while coarse was the sum of 6 km, and 12 km. The six model predictors were tested for collinearity using vifstep() in the *usdm* R package (Naimi et al., 2014) with a variance inflation factor threshold of 6, and no problems were found. Linear models were built using the lm() R function. All predictors were initially included, then using the step() function, pared down in each

regression model using stepwise bidirectional selection based on the Akaike information criterion, using 4 degrees of freedom as a penalty to make predictor selection stricter than the default. The resulting regression model was used to calculate a climate value at each grid cell based on a linear relationship with the selected predictors.

Second, to adjust for local variation in climate that is not associated with topography, the linear model residuals at each station were calculated and interpolated using the thin plate spline approach implemented in the *fields* R package (Nychka et

al., 2017). The lambda smoothing parameter, which determines how closely the fitted surface matches input values, was set to 0.01. This low lambda value was selected because of the relatively high confidence in the long-term averaged weather station values (based on at least 8 years of data). This effectively produces a smoothed layer of local deviation from the linear model, which was used to additively adjust the results of the linear model predictions and produce finalized climate rasters.

We measured the spatial predictive ability of models using ten-fold cross-validation (Dobesch et al., 2007). In each validation round, 10% of weather stations were reserved as a test dataset and the remainder were used for training. While randomly selected test points may be subject to spatial sampling bias (Hijmans, 2012), this may be less of a concern for this study because in Hong Kong the weather stations are fairly stratified (Figure S2). Average root mean squared error of the test data subset from the final model prediction was used as an error measurement. To normalize these error measures across the climate

variables, we adjusted them as a percentage of the standard deviation of the initial weather station values measured. This cross-validation procedure was used only to produce these validation measurements. The finalized monthly climate rasters described above were trained using all available data.

The finalized monthly rasters were then summarized into layers that characterize yearly climatic means and variation. These include 19 "bioclimatic" variables using the biovars() function in the *dismo* R package (Hijmans et al., 2017), which are

specifically suited for species distribution modeling and other ecological purposes. This also allows our data to be compared with other climate data products that use the same calculations. Because those calculations only use rainfall and average daily maximum and minimum temperatures in each month, we also produced yearly average layers of dewpoint, relative humidity, mean daily temperature, air pressure, and wind speed. Also provided are layers of highest and lowest average monthly extreme temperatures, and their difference (extreme temperature annual range). Because they are derived from monthly extremes rather

than averaged daily extremes, these variables represent the full range of temperatures experienced in a given location better than the bioclimatic variables.

For comparison with global climate data products, we resampled bioclimatic variables to the same (1 km) resolution as WorldClim using bilinear interpolation. Only pixels present in both data products were used for comparisons.

### 3.3   Remote sensing data

Normalized difference vegetation index (NDVI) is a common metric of vegetation presence and density derived from satellite imagery. To calculate NDVI, Landsat 8 images (U.S. Geological Survey, 2018) of Hong Kong were obtained. We downloaded one image from March 2016 that covers much of Hong Kong except for the far eastern areas, and is free of clouds. This was supplemented with an image from March 2018 after adjustment, so that all land areas of the region were included. NDVI calculations were completed using the standard equation (Pettorelli et al., 2005):

$NDVI = (NIR - Red)/(NIR + Red)$  (1)

Where NIR is near-infrared (Landsat band 5: 0.851 to 0.879 μm) and Red is visible red radiation (Landsat band 4: 0.636 to 0.673 μm). The resulting NDVI value varies between 1 and -1, where higher values correspond with denser vegetation.

## 4  Results and discussion

Results of this environmental analysis of Hong Kong include 48 rasters and one vector file. All rasters are provided at an identical 1 arc second (30 m) resolution and in the WGS84 geographic coordinate system. Summary values and filenames are provided in the data repository.

### 4.1  Topographic variables

Distance to coast results show that approximately 42% of Hong Kong's land area is within 1 km of the coastline. However it is apparent that inland areas often feature steep inclines, as half of Hong Kong's land is above 84 m elevation.

For variables like relative elevation, urbanicity, and water proximity, the ideal scale of raster calculation is dependent on the desired effect to be captured, and perhaps other characteristics of the landscape in question. For this reason, we provide these rasters calculated at multiple buffer scales.

Urbanicity results show that the majority of land in Hong Kong is not near urban areas, as the median raster values are below 4% urban at all scales calculated (Table 1). This shows that although Hong Kong has extremely dense urban cores, most of its mountainous terrain is unpopulated.

### 4.2  Climate variables

Minimally, a total of 32,024 monthly weather station measurements over 20 years (1998 to 2017) were used to construct climate models for all months and variables, at finer resolution compared to global datasets (Fig. 2). High weather station density and availability of data on multiple candidate topographic climate-forcing factors allowed for high confidence in many climate variable models, especially those related to temperature (Figs. 3, 4). The climate interpolation results include monthly models of ten variables including temperature, precipitation, and humidity, making a total of 120 individual models produced (monthly models of three temperature variables are shown in Fig. 5). As an example, one of these models represents minimum temperatures recorded in all Januaries with data available from 1998 to 2017. For all variables, the predictors included in monthly models are displayed in Figure 6, and the number of stations with data included is in Table 2.

#### 4.2.1  Temperature

Temperature was found to vary considerably across Hong Kong, with more than 6ºC difference in mean annual temperature between the highest mountain peaks (>900 m, <18ºC) and some low-lying urbanized areas (>24ºC). While mean and minimum temperature are highest in urban areas, maximum temperature shows a different pattern with a maximum in inland valleys in the northern New Territories. This pattern may be explained by urban heat retention: buildings act as heat sinks which absorb solar radiation during the day, and slowly release heat at night, causing increased minimum temperatures (see Oke, 1982). The high maximum temperatures in inland valleys may be due to reduced air circulation in sheltered locations, and lack of complex vegetation or urban structures providing shade. The high accuracy of temperature models (Figs. 3, 4) is likely due to a strong association with elevation; elevation was by far the most commonly included predictor for temperature models (Fig.

6). Urbanicity was important for mean and minimum temperature, but not maximum temperature. Water proximity and coast distance were differentially included depending on the variable, while aspect*slope rarely had an effect.

### 4.2.2 Rainfall

In our models, the highest annual rainfall (bio12) areas in Hong Kong (>2500 mm annually) are inland and at high elevations,
presumably because of condensation from humid air as it passes over mountains. Areas near the coast, particularly small outlying islands and the eastern coast in Lung Kwu Tan receive the lowest amount of annual rainfall (<1600 mm). Precipitation of driest month (bio14) was uniformly low, ranging from 20 to 40 mm, but the relative pattern of high and low precipitation areas remained similar. The most commonly included model predictor was fine-scale water proximity (Figure 6). Elevation was predictive for 5 out of 12 months, but few other topographic predictors were useful. Seasonality of rainfall in Hong Kong is
strong. Averaged across all locations, 52% of total yearly rainfall was recorded in three months (June through August). Rainfall models were informed by more weather stations than any other climate variable (Table 2), but they have the highest relative standard error (Fig. 3) and therefore the lowest accuracy. Because they are influenced by both global and locally variable wind patterns, precipitation distributions are notoriously difficult to predict, especially in urban areas (Cristiano et al., 2017).

### 4.2.3 Dew point, humidity, pressure, and wind speed

Dew point exhibits a similar pattern to other temperature variables, with mean annual dew point ranging from 15.5ºC at mountain peaks to around 19ºC on small islands and lower areas. Mean annual relative humidity reaches a maximum of about 90% at Tai Mo Shan, while many urban areas in Kowloon, Tuen Mun, and Yuen Long are between 70 and 75%. Surprisingly, mean annual air pressure has a positive correlation with elevation; the highest values (reaching 1014 hPa) are at mountain peaks, and particularly low values (as low as 1012.5 hPa) in coastal areas of southern and western Hong Kong. Mean annual
wind speed is also strongly associated with elevation, with mean annual values above 30 km/h on Lantau Island mountain peaks, down to below 5 km/h in interior low elevation areas of the New Territories.

### 4.2.4 Comparisons with global climate data

Our new climate models are compared with a recent global climate dataset to identify differences in predictions of Hong Kong climate values (Fig. 7). WorldClim 2 was produced using a similar interpolation approach with regression modeling and thin
plate spline interpolation, but also included satellite-derived covariates in addition to topography (Fick and Hijmans, 2017). Because WorldClim incorporates vast amounts of data from multiple databases covering overlapping geographic and political entities, it is difficult to ascertain exactly which individual weather stations were included, and we were unable to determine whether any Hong Kong weather stations were included or if the datasets are completely independent. However, the model predictions differ substantially (Figs. 2, 7; Table 3). Our models generally indicate greater spatial variation than WorldClim,
with cool areas colder, warm areas hotter, and wet areas wetter. For example in average low temperature of coldest month (bio6), high elevation areas could be more than 2ºC lower, and urban areas more than 2ºC higher than WorldClim indicates (Fig. 7a).

To further quantify differences in values between these two datasets, for each of the 19 bioclimatic variables we calculated the standard deviation of raster values (Table 3). All of our interpolated climate rasters had a higher standard deviation than their WorldClim 2 counterparts. Though there is a temporal discrepancy between weather station data used in WorldClim 2 (1970-2000) and this study (1998-2017), climate change is unlikely to explain the observed differences in temperature variability.

Evidence suggests that if anything, mountains are experiencing climate warming faster than low elevation areas (Pepin et al., 2015), which would give the opposite results of our findings where Hong Kong's mountains are cooler than WorldClim indicates (Fig. 7a). Unless global climate models increase in resolution and accuracy, regional models will remain critical for local applications.

## 4.3  Remote sensing variable

The NDVI data represents vegetation quality and density based on two merged satellite images, both in March of their respective years. Although this is only an instantaneous representation of NDVI, we expect it to correlate strongly with the spatial pattern of vegetation density throughout the year. Certain plant species shed and regenerate their leaves during specific months ranging from winter through mid-summer, but Hong Kong's woody vegetation is overall evergreen (Dudgeon and Corlett, 1994), so seasonal changes in NDVI are not expected to be drastic. NDVI values above 0.4 include Hong Kong's densest

forests, while unvegetated or urbanized areas are well below 0.1. The densest vegetation (> 0.4 NDVI) in Hong Kong tends to be on slopes between 100 m and 400 m elevation (Fig. 8), and is distributed between Hong Kong Island, Lantau Island, and the New Territories. The verdant mangrove forests, at sea level, are an exception. The patchy distribution of high density vegetation likely reflects the effects of historical deforestation. The largest patches are found on the southeastern slopes of Tai To Yan in the New Territories. The relative distribution of NDVI classes along Hong Kong's elevational gradient is shown in Figure 8.

Future work could determine to what extent NDVI changes over time, in response to seasonality or recent weather. The limiting factor is the availability of data of adequate temporal resolution, as many satellite images of Hong Kong are obscured by cloud cover or degraded by poor air quality.

## 4.4  Value and Utility

This new data will benefit environmental research, and specifically SDM studies, in two main ways. First, it will enable

finer scale analyses than previously possible. For SDM, this means improved detection of climatic microrefugia (Meineri and Hylander, 2017), and the ability to differentiate between human altered habitat and natural areas. Rampant development and a shifting climate make this knowledge of local species persistence more important than ever. Additionally, this is especially relevant in Hong Kong where topography varies dramatically, and where urban areas form a complex mosaic with undeveloped expanses.

Second, we provide a diverse array of rasters derived from multiple independent data sources, but in a single resolution and format to facilitate further analysis and synthesis of meaning. For SDM, these layers have distinct advantages over datasets that only contain climate data. Compared to climate data alone, using diverse predictors including topographic characteristics have been shown to be important variables for accurate SDM results, such as predicting the spread of invasive species in new

ranges (Peterson and Nakazasa, 2008). However benefits of non-climate data may only be evident in finer scale SDMs (Luoto et al., 2007).

Finally, such high quality, diverse geographic data is especially uncommon in tropical regions, where improved knowledge for environmental research and biological conservation is most needed. According to Rapoport's Rule, tropical species are more likely to have smaller distributions (Stevens, 1989), and therefore future execution of local SDM studies to understand their ranges are particularly important.

## 4.5 Limitations and next steps

Here we outline how shortfalls of the presented data may be improved in the future. First, though we inferred Hong Kong's pattern of urban development from impervious surface data, this is less than ideal because in addition to concrete, bare soil or rock are sensed as impervious. Also, it cannot differentiate dense urban cores of high-rises from large paved areas. For climate modeling, an urbanicity measure that considers building height or population density at a 30 m or finer scale could be preferable.

Second, while our temperature rasters should accurately represent air temperature in open areas, they do not reflect the high spatial variation in temperature found in urban microclimates. For example, although the manned Kowloon HKO weather station is inside a densely populated area, as pointed out by Nichol and To (2012) it is still in a small parklike area surrounded by trees, and therefore is not representative of the most densely urbanized areas of Hong Kong. Other stations in urban areas are similarly near green spaces or otherwise open areas. Higher resolution (say 5 m or 1 m) studies of urban thermal distributions would strongly benefit from analysis of wind patterns, building height, thermal pollution, and other factors (e.g. Shi et al., 2018). Therefore granular, ground-level temperatures in urban areas are likely substantially different than the broader air temperature values our models provide.

Similar to other climate interpolation studies, bias in the physical locations of automatic weather stations may be of concern. Weather stations are often intentionally placed in flat, open areas with the goal of measuring weather that is relevant to a broad geographic area, rather than locations that may experience unique local climate. It may be for this reason that Slope*Aspect was infrequently useful for model construction, as few stations are on steep slopes. Elevational distribution of stations may also be a source of bias; although a weather station operates at the highest point in Hong Kong (Tai Mo Shan, 955 m), there are only two other stations above 600 m.

Finally, while we used cross-validation to measure the spatial predictive ability of the climate models, this method is only able to test models against locations where weather stations are present; validation based on an independently collected dataset would be ideal. One common validation method is to use weather data loggers placed across elevational and land-use gradients (Meineri and Hylander, 2017). Such an approach would allow for explicit testing and comparing predictiveness of climate products for different areas of Hong Kong.

Important gaps in Hong Kong geographic data remain. Projections of future climate scenarios could complement historical data to enable predictions of biodiversity change. Additional variables like cloud cover and solar radiation would especially benefit studies of photosynthetic taxa. A discrete classification of habitat type would be useful for ecological research, and

quality soil type data is lacking. Availability of such data for Hong Kong would complement the findings of this project, which significantly advance our understanding of geographic heterogeneity in this complex tropical region.

## 5   Conclusions

This diverse set of 30 m resolution topography, climate, and remote sensing data include the first published interpolation of
long-term climate averages specific to Hong Kong. Our findings suggest that global interpolated climate datasets are limited by their resolution, and underestimate local climate variability. Therefore the availability of such local data will remain critically important for the foreseeable future. This new data will allow for a new generation of studies in Hong Kong, and enable connections between environmental data and biotic patterns at a much finer scale than previously possible. Aside from clear uses in conservation, ecological and biogeographic research, we also expect this freely accessible dataset to be broadly applicable
for many sectors, including tourism, hydrology, recreation, agriculture, mapmaking, and real estate.

## 6   Data availability

GeoTIFF raster and shapefile documents (Morgan and Guénard, 2018) can be downloaded from figshare: https://doi.org/10.6084/m9.figshar A document in the repository includes file names, descriptions, and summary statistics for all provided rasters. Individual monthly rasters for each of the 10 climate variables are available as a compressed zip file.

*Author contributions.* BAM acquired initial data, conducted modeling, and prepared the dataset. BAM and BG prepared the manuscript.

*Competing interests.* The authors declare that they have no conflict of interest.

*Acknowledgements.* We thank Ocean Park Conservation Foundation for supporting this research. This project would not have been possible without the Hong Kong Observatory, which works tirelessly to maintain their weather station network and ensure the resulting data is accessible. We also thank Eric Meineri for comments and advice while planning our analyses.

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

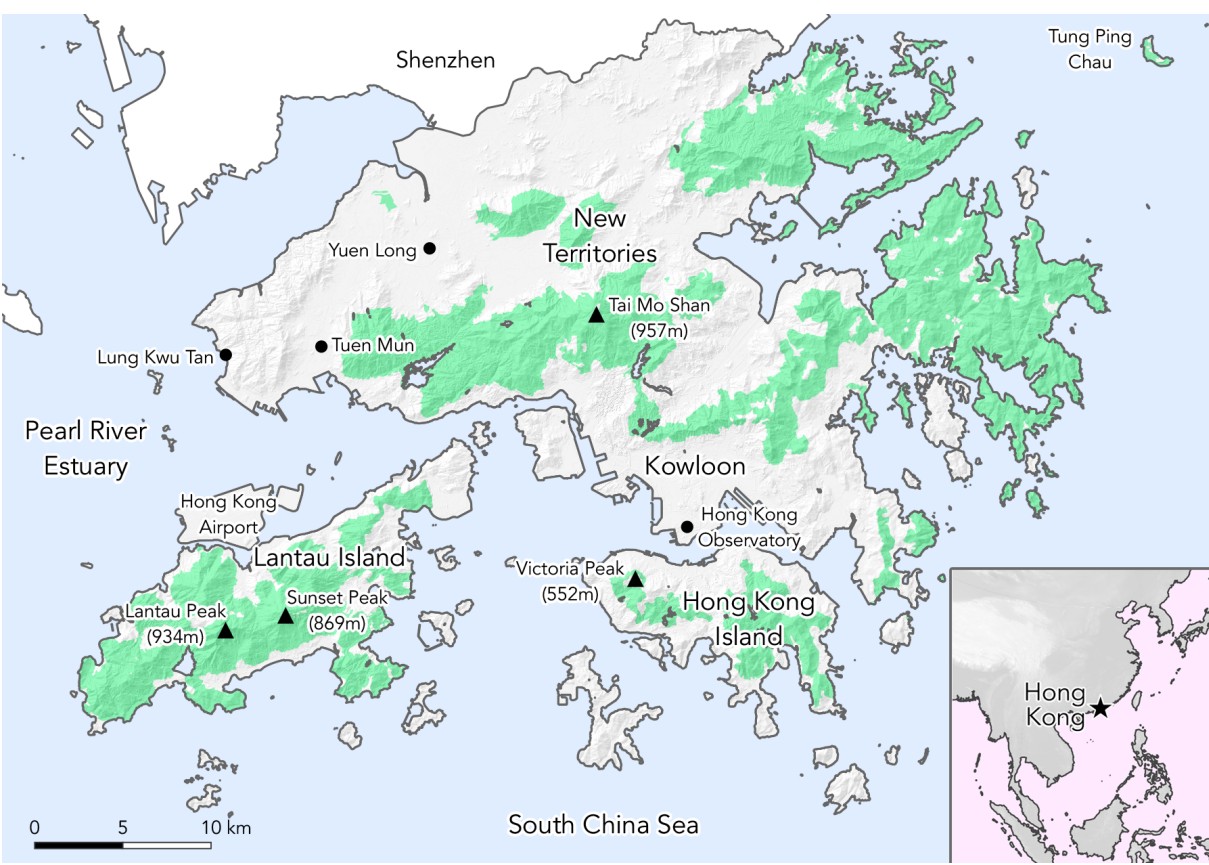

**Figure 1.** Hong Kong geography. The three highest peaks in the territory, as well as the highest point on Hong Kong Island are marked. Areas protected as Country Parks are highlighted in green.

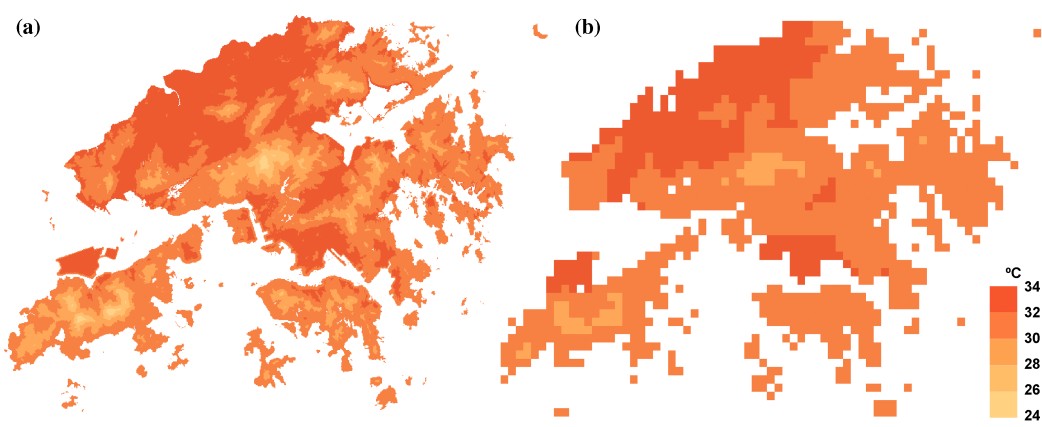

**Figure 2.** Comparison of average high of warmest month (bio5) model results for Hong Kong. (a) is from our newly interpolated climate models at 30 m resolution, while (b) is 1 km resolution data available as part of WorldClim 2 (Fick and Hijmans, 2017). Not only is the resolution markedly improved, but also the temperature values are more varied, for instance on the large southern islands.

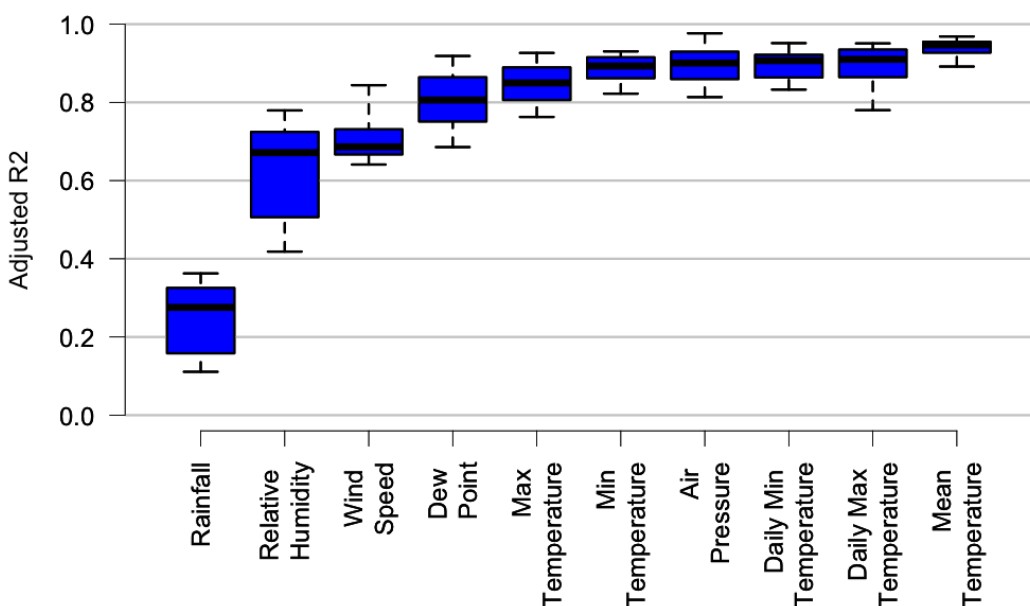

**Figure 3.** Adjusted r2 values of initial (pre-spline) regression models. Each boxplot includes 12 points, one for each monthly model. Temperature variation, especially mean temperature, was best explained by linear modeling, while rainfall was predicted the most poorly.

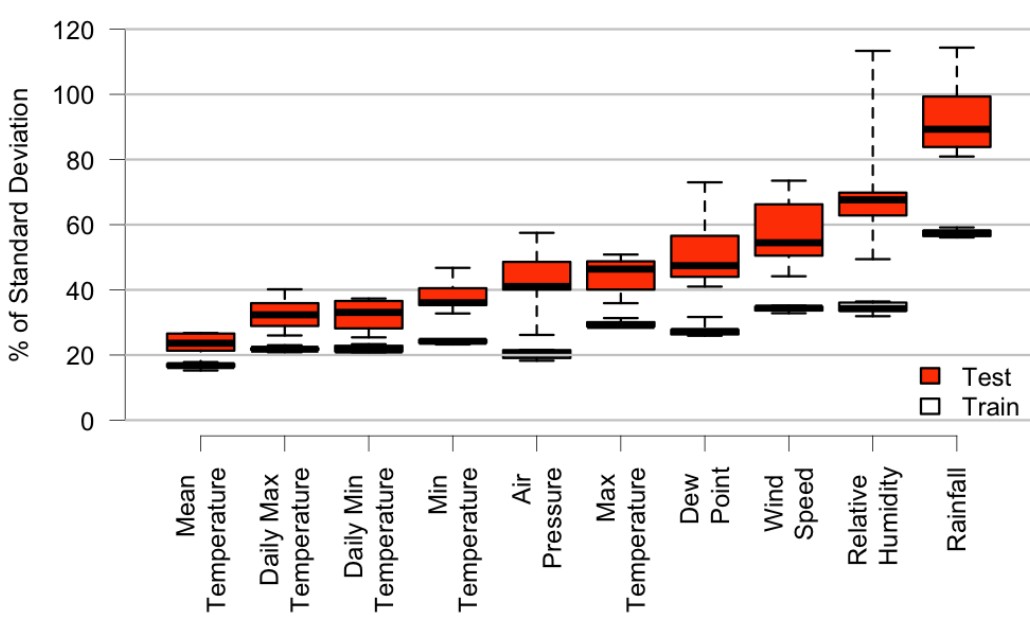

**Figure 4.** Relative magnitude of training and testing dataset errors, from 10 validation rounds of climate variable modeling. A value of 100 indicates for that climate model, that the average difference between the value recorded at a given weather station and the value predicted by the model at that location, is equal to the standard deviation of the initial set of all values recorded at all weather stations for that climate variable.

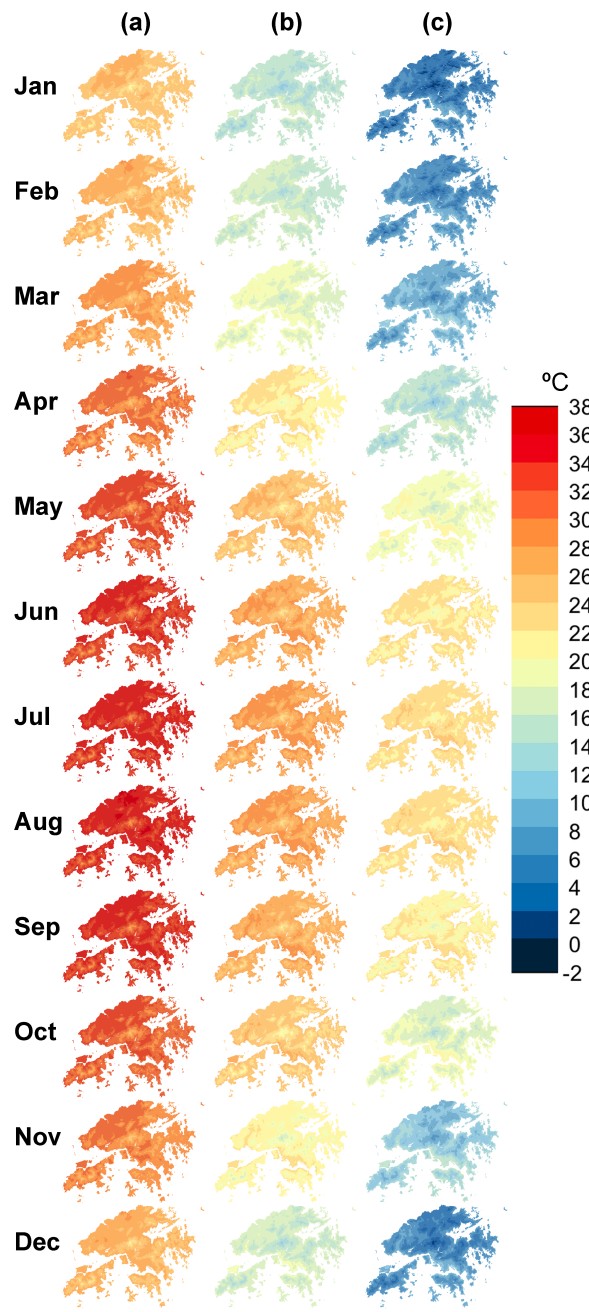

**Figure 5.** Model results for three of ten interpolated climate variables. (a) Maximum temperature, (b) Mean temperature, and (c) Minimum temperature.

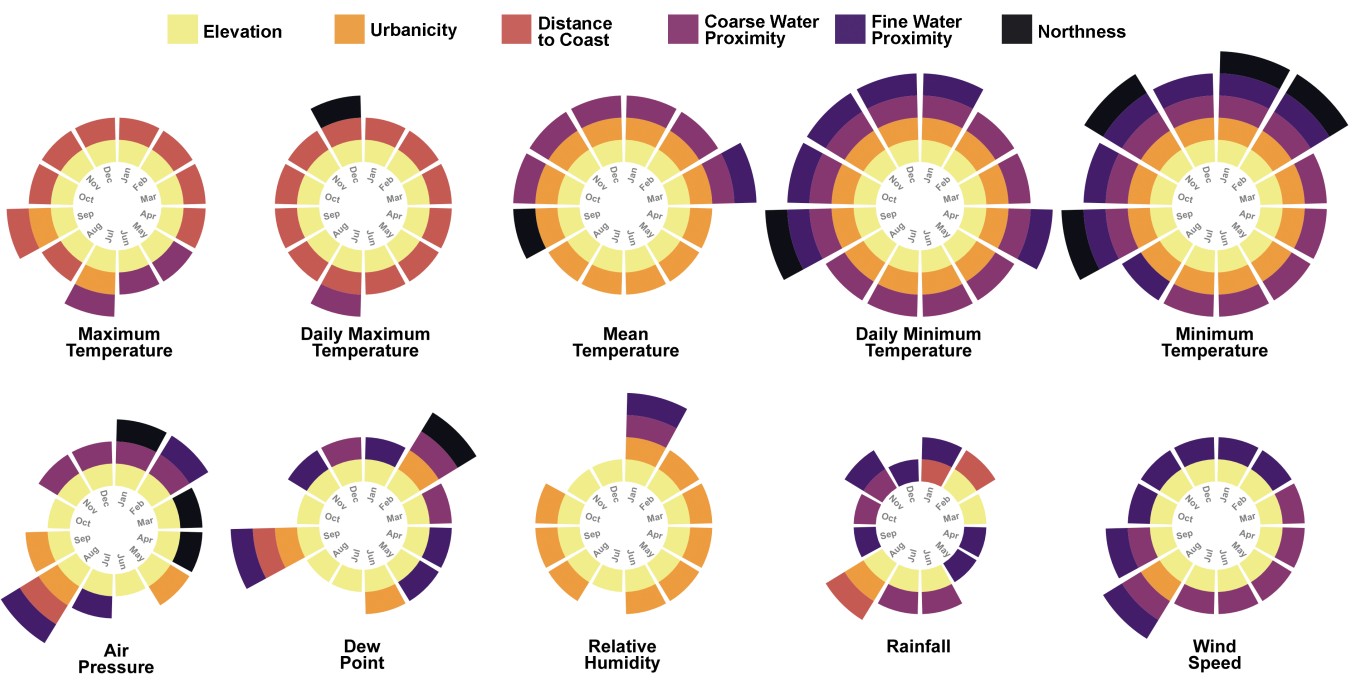

**Figure 6.** Regression predictors included in monthly models for 10 climate variables. Each predictor is represented by a different color. Minimum and mean temperature variables were most predictable, consistently including elevation and urbanicity. Rainfall patterns were most difficult, with the fewest predictors included.

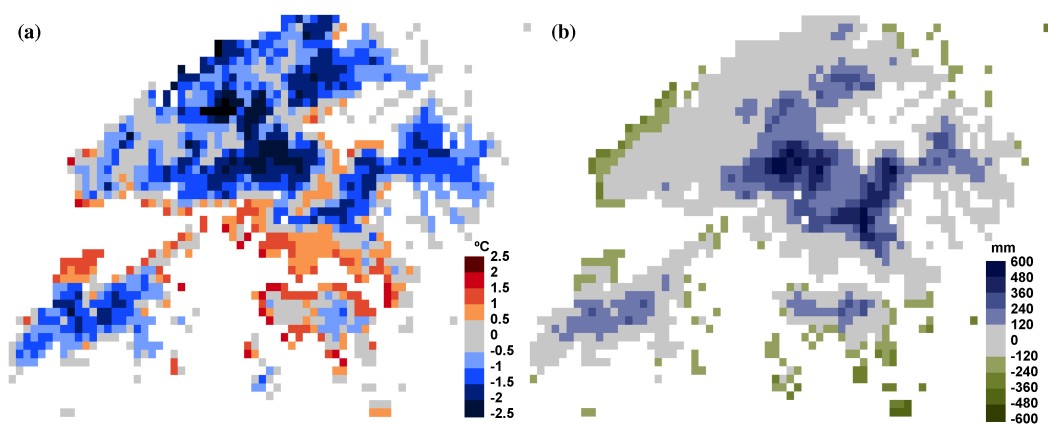

**Figure 7.** Differences between results of this study and Worldclim 2 (Fick and Hijmans, 2017) values. (a) is average low temperature of coldest month (bio6), with red where the local model is warmer than WorldClim, and blue is colder. (b) shows annual precipitation (bio12), with blue where the local model predicts more rainfall than WorldClim, and tan is less rainfall. Our model results were resampled to 1 km resolution using bilinear interpolation to allow for these comparisons.

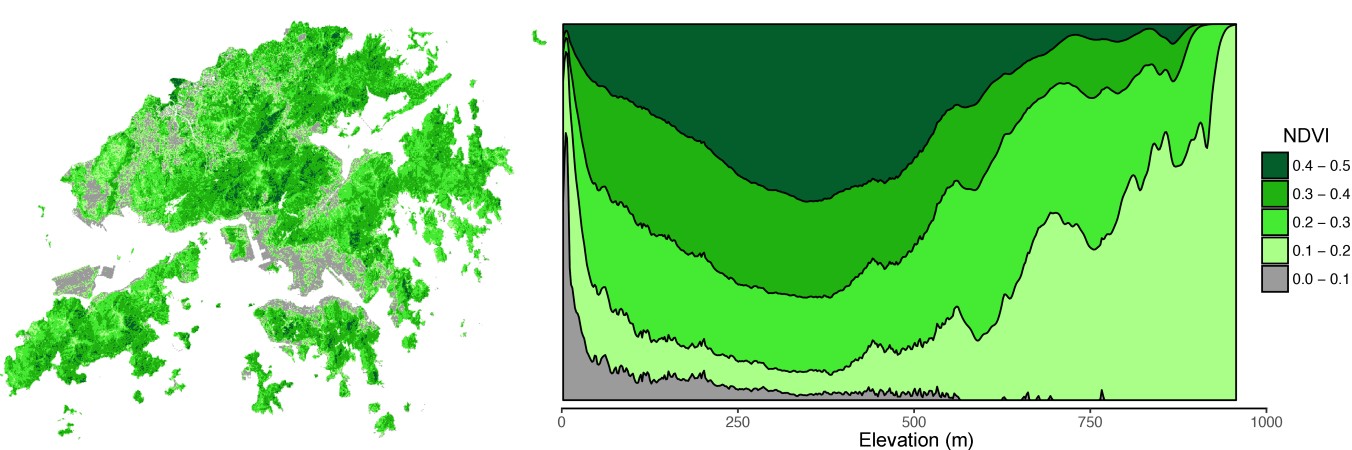

**Figure 8.** NDVI class composition over Hong Kong's elevational range. The majority of land area near sea level is below NDVI 0.1, while Hong Kong's highest elevation areas are between 0.1 and 0.2, indicating short vegetation. The elevation range with proportionally the most dense vegetation (0.4 to 0.5 NDVI) is 300 to 400 m.

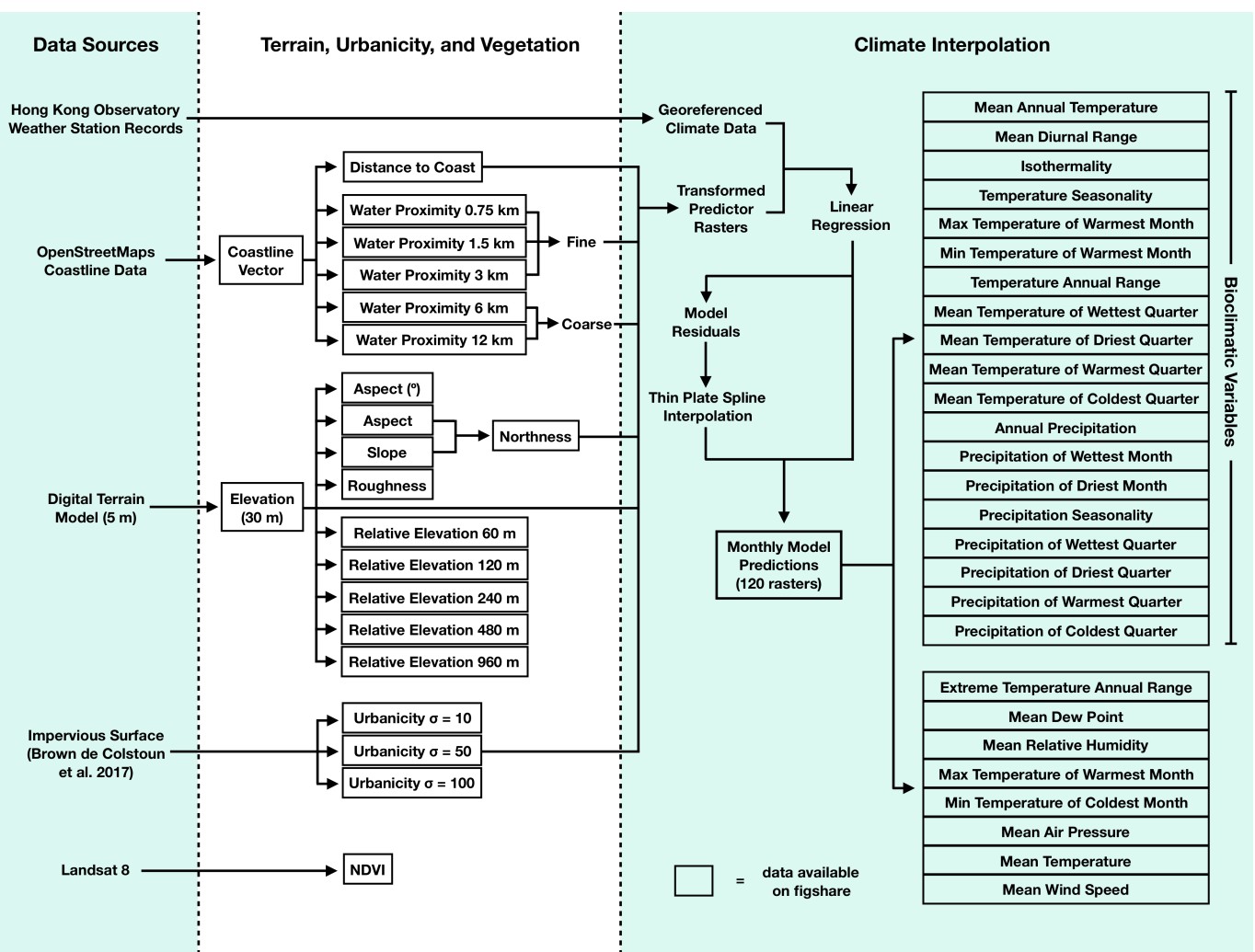

**Figure S1.** Schematic of data products and the sources that informed them. Items enclosed in a box represent the files available for download from the figshare repository.

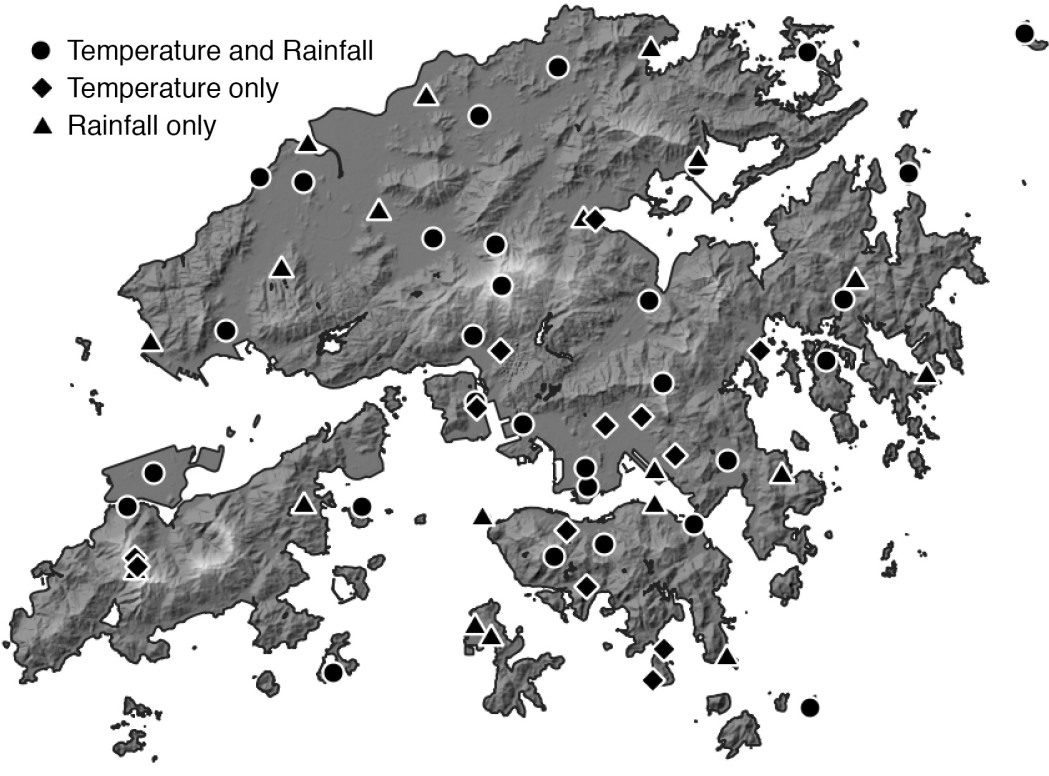

**Figure S2.** Permanent weather stations operated by the Hong Kong Observatory. Symbols indicate what type of data is available from each station: temperature, rainfall, or both.

**Table 1.** Raster product descriptions, units, and 5th, 50th, and 95th percentile values.

| Description | Unit | 5% | 50% | 95% | Filename |
|---|---|---|---|---|---|
| Aspect (Northness) | index | -0.99 | 0 | 0.99 | aspect.tif |
| Aspect (Degree) | º | 18 | 180 | 341 | aspect_degree.tif |
| Slope | º | 0 | 17 | 33 | slope.tif |
| Terrain Roughness | index | 0.33 | 24.95 | 50.67 | rough.tif |
| Elevation | m | 5 | 84 | 407 | elevation.tif |
| Aspect * Slope | index | -23.5 | 0 | 23.58 | aspect_x_slope.tif |
| Distance to Coast | m | 68 | 1349 | 6186 | waterdist.tif |
| Relative Elevation (60 m radius) | m | 0 | 16 | 37 | relelev60.tif |
| Relative Elevation (120 m radius) | m | 0 | 28 | 69 | relelev120.tif |
| Relative Elevation (240 m radius) | m | 2 | 46 | 124 | relelev240.tif |
| Relative Elevation (480 m radius) | m | 2 | 64 | 208 | relelev480.tif |
| Relative Elevation (960 m radius) | m | 3 | 76 | 308 | relelev960.tif |
| Water Proximity (0.75 km radius) | proportion | 0.52 | 1 | 1 | water25.tif |
| Water Proximity (1.5 km radius) | proportion | 0.4 | 0.98 | 1 | water50.tif |
| Water Proximity (3 km radius) | proportion | 0.33 | 0.88 | 1 | water100.tif |
| Water Proximity (6 km radius) | proportion | 0.31 | 0.74 | 1 | water200.tif |
| Water Proximity (12 km radius) | proportion | 0.27 | 0.66 | 0.94 | water400.tif |
| Annual Mean Temperature | ºC | 20.8 | 22.9 | 24 | biovars_t_1.tif |
| Mean Diurnal Range (Mean (max temp-min temp)) | ºC | 4.9 | 6.2 | 7.7 | biovars_t_2.tif |
| Isothermality (bio2/bio7) (* 100) | index | 27.4 | 31.9 | 35.6 | biovars_t_3.tif |
| Temperature Seasonality (standard deviation *100) | index | 467 | 496 | 512 | biovars_t_4.tif |
| Average High Temperature of Warmest Month | ºC | 28.9 | 31.5 | 32.8 | biovars_t_5.tif |
| Average Low Temperature of Coldest Month | ºC | 9.5 | 11.7 | 13.9 | biovars_t_6.tif |
| Temperature Annual Range (bio5-bio6) | ºC | 17.7 | 19.6 | 21.6 | biovars_t_7.tif |
| Mean Temperature of Wettest Quarter | ºC | 25.8 | 27.8 | 29.2 | biovars_t_8.tif |
| Mean Temperature of Driest Quarter | ºC | 14.4 | 16.3 | 17.4 | biovars_t_9.tif |
| Mean Temperature of Warmest Quarter | ºC | 25.9 | 28.2 | 29.2 | biovars_t_10.tif |
| Mean Temperature of Coldest Quarter | ºC | 14.4 | 16.3 | 17.4 | biovars_t_11.tif |
| Annual Precipitation | mm | 1738 | 2079 | 2415 | biovars_t_12.tif |
| Precipitation of Wettest Month | mm | 345 | 425 | 521 | biovars_t_13.tif |
| Precipitation of Driest Month | mm | 25 | 32 | 35 | biovars_t_14.tif |
| Precipitation Seasonality (Coefficient of Variation) | index | 78.7 | 82.8 | 86 | biovars_t_15.tif |

| | | | | | |
|---|---|---|---|---|---|
| Precipitation of Wettest Quarter | mm | 883 | 1085 | 1276 | biovars_t_16.tif |
| Precipitation of Driest Quarter | mm | 86 | 104 | 112 | biovars_t_17.tif |
| Precipitation of Warmest Quarter | mm | 814 | 1054 | 1260 | biovars_t_18.tif |
| Precipitation of Coldest Quarter | mm | 86 | 104 | 112 | biovars_t_19.tif |
| Extreme Temperature Annual Range | ℃ | 26.3 | 29 | 32.1 | avars_annual_range.tif |
| Annual Mean Dew Point | ℃ | 17.3 | 18.4 | 19.1 | avars_dewp_mean.tif |
| Annual Mean Relative Humidity | % | 75.4 | 80.4 | 84.9 | avars_humid_mean.tif |
| Maximum Temperature of Warmest Month | ℃ | 32.3 | 35 | 36.2 | avars_max_tmax.tif |
| Minimum Temperature of Coldest Month | ℃ | 2.4 | 5.6 | 8.6 | avars_min_tmin.tif |
| Annual Mean Air Pressure | hPa | 1012.5 | 1012.8 | 1013.4 | avars_press_mean.tif |
| Actual Annual Mean Temperature | ℃ | 20.3 | 22.4 | 23.6 | avars_tmean_mean.tif |
| Annual Mean Wind Speed | km/h | 5.4 | 11.6 | 19.2 | avars_windsp_mean.tif |
| Urbanicity (sigma = 10) | % | 0 | 0 | 68.9 | urbanicity_gauss10.tif |
| Urbanicity (sigma = 50) | % | 0 | 1.5 | 56 | urbanicity_gauss50.tif |
| Urbanicity (sigma = 100) | % | 0 | 3.3 | 50.1 | urbanicity_gauss100.tif |
| Normalized Difference Vegetation Index (NDVI) | index | 0.05 | 0.29 | 0.39 | hk_ndvi.tif |
| Hong Kong Coastline and Reservoirs | - | - | - | - | HK_border.shp |

**Table 2.** Number of weather stations that contributed data for each climate model.

|  | press | tmax | mtmax | tmean | mtmin | tmin | dewp | humid | prec | windsp |
|---|---|---|---|---|---|---|---|---|---|---|
| Jan | 17 | 39 | 39 | 38 | 39 | 38 | 23 | 23 | 40 | 28 |
| Feb | 17 | 40 | 40 | 39 | 40 | 39 | 25 | 25 | 41 | 28 |
| Mar | 17 | 39 | 39 | 38 | 39 | 39 | 25 | 25 | 40 | 28 |
| Apr | 18 | 39 | 39 | 37 | 39 | 39 | 24 | 24 | 41 | 29 |
| May | 17 | 39 | 39 | 39 | 39 | 39 | 24 | 24 | 41 | 27 |
| Jun | 16 | 38 | 38 | 37 | 38 | 38 | 24 | 24 | 42 | 27 |
| Jul | 17 | 37 | 37 | 37 | 37 | 37 | 24 | 24 | 41 | 28 |
| Aug | 17 | 39 | 39 | 39 | 39 | 39 | 25 | 25 | 40 | 27 |
| Sep | 16 | 40 | 40 | 38 | 40 | 40 | 25 | 25 | 41 | 27 |
| Oct | 18 | 42 | 42 | 42 | 42 | 42 | 26 | 26 | 43 | 29 |
| Nov | 18 | 42 | 42 | 41 | 42 | 42 | 26 | 26 | 43 | 29 |
| Dec | 18 | 43 | 43 | 42 | 43 | 42 | 25 | 25 | 44 | 29 |

**Table 3.** Comparisons of variation between bioclimatic variables, measured as raster value standard deviation. All new rasters are more variable than their corresponding Worldclim 2 layers. Increases in standard deviation range from 1.4x to 3.4x. Calculations may appear inaccurate due to rounding.

|        | Local Model SD | Worldclim 2 SD | Increase Ratio |
|--------|----------------|----------------|----------------|
| bio 1  | 1.0            | 0.5            | 1.9            |
| bio 2  | 0.8            | 0.3            | 3.0            |
| bio 3  | 2.5            | 0.7            | 3.4            |
| bio 4  | 14.6           | 10.2           | 1.4            |
| bio 5  | 1.2            | 0.7            | 1.7            |
| bio 6  | 1.3            | 0.5            | 2.8            |
| bio 7  | 1.2            | 0.6            | 2.0            |
| bio 8  | 1.1            | 0.6            | 1.8            |
| bio 9  | 0.9            | 0.5            | 1.9            |
| bio 10 | 1.1            | 0.6            | 1.7            |
| bio 11 | 0.9            | 0.5            | 1.9            |
| bio 12 | 204.4          | 95.4           | 2.1            |
| bio 13 | 52.9           | 21.5           | 2.5            |
| bio 14 | 3.1            | 1.6            | 1.9            |
| bio 15 | 2.2            | 1.1            | 2.0            |
| bio 16 | 119.9          | 54.2           | 2.2            |
| bio 17 | 8.2            | 4.1            | 2.0            |
| bio 18 | 136.2          | 67.9           | 2.0            |
| bio 19 | 8.2            | 5.5            | 1.5            |

**Appendix A: Glossary of variable definitions**

**Maximum temperature (tmax)**    the highest temperature observed within a month

**Mean daily maximum temperature (mtmax)**    the mean of all daily high temperatures within a month

**Mean daily temperature (tmean)**    the mean of all temperatures within a month

**Mean daily minimum temperature (mtmin)**    the mean of all daily low temperatures within a month

**Minimum temperature (tmin)**    the lowest temperature observed within a month

**Mean dew point (dewp)**    the mean of all dew point observations within a month

**Mean relative humidity (humid)**    the mean of all relative humidity observations within a month

**Mean wind speed (windsp)**    the mean of all wind speed observations within a month

**Mean air pressure (press)**    the mean of all air pressure observations within a month

**Rainfall (prec)**    the total of all rain recorded within a month

**Relative elevation**    the difference in elevation between the pixel of interest, and the lowest pixel within a given radius

**Distance to coast**    geometric distance between the pixel of interest and the nearest oceanic coastline

**Water proximity**    percent of area that is terrestrial within a given radius of the pixel of interest

**NDVI**    Normalized Difference Vegetation Index

**Urbanicity**    measure of area that is impervious surface within a given radius of the pixel of interest