# Peer review of "New 30 m resolution Hong Kong climate, vegetation, and topography rasters indicate greater spatial variation than global grids within an urban mosaic"

_Earth System Science Data, 2018_

## Referee Comment (RC1) · Anonymous Referee #1 · 12 Jan 2019

General comments

The manuscript is not sufficiently organized and confused with no novelty and explicit research question. There are many too short subsections, which should be merged. Methods are not much clear because details and relevant references have not been provided. Consequently, it is not much easy to follow results and discussion. The Authors have used data associated at support sizes very different. They should take into account the change of support.

Detailed comments

The title should be made more informative and effective. The Abstract has not the required structure and does not summarize the whole manuscript. It should be organized better and explain clearly what was done, what was found and what are the main conclusions. Generally, the first sentence should provide briefly the rational of the topic being investigated.

Keywords are missing.

The Introduction section is confused ant not sufficiently organized. Particularly, reading the title, one is expecting to find in the Introduction the presentation of what the title promises, but unfortunately it is not so. The Introduction should be improved and the topic being investigated should be explained clearly. The novelty and objectives are missing. A manuscript to be considered a research paper, a research question must be clearly stated. In addition, the Authors should explain the gap in the topic being investigated and how their study fills such a gap.

A well-organized Materials and Methods section is missing. The sections '2 Study area' and '3 Methods' should be included in a new Materials and Methods section which allows readers to follow the progress of the objectives in the manuscript and support results and discussion. In the methods, how data have been analysed and combined should be explained providing sufficient details. Particularly, the Authors should explain how they have taken into account the change of support problem to have all data associated to the same support size. Details and references on statistical methods are missing.

Results and Discussion sections should be improved and supported by a new Materials and Methods section.

Conclusions are poor: they should be improved and to show the improvement of our knowledge.

---

## Referee Comment (RC2) · Anonymous Referee #2 · 22 Jan 2019

This study aims to produce a high resolution (30 m raster) data set of climate and environmental variables for the Hong Kong region. Unfortunately, I find the manuscript to be confusing, showing an overall disconnection between sections. The manuscript focuses on a large but incomplete description of the variables included in the data set, and does not address the main conclusion stated in its title ("Local models reveal greater spatial variations than global grid in an urban mosaic").

The introduction section discusses the application of "Species distribution modeling (SDM)" and how this type of analysis is affected by the spatial resolution of the envi-

ronmental data employed. However, this introductory discussion seems to be irrelevant within the context of the manuscript, as SDM is rarely mentioned again throughout the text. Abbreviations such as NDVI are used throughout the abstract and introduction but are not explained until the later sections of the methods section.

In the method section each of the topographic and climate variables, as well as remote sensing products are mentioned. However, it seems to me that each of the subsections focuses on irrelevant details, and there is no clear descriptive explanation of a) what these variables are? b) why were they chosen? and c) how were they processed?

The results and discussion section is also vague and difficult to read. There is no clear distinction between the validation data set/model and the novel data/model analysis produced by this study. The figures lack explanation within the main text, and it is hard to see how they convey the results of the study.

Overall, I believe this manuscript needs substantial revisions, and perhaps a reassessment of the scientific goals that it is trying to communicate.

———————————————

---

## Referee Comment (RC3) · Anonymous Referee #3 · 23 Jan 2019

Review of Morgan and Guenard ESSD Jan 2018

This paper describes a dataset of geographic, biological and meteorological rasters at 30 m resolution for the territory of Hong Kong.

It is clear how and why these data are useful, and the paper describes the process used to create the dataset fairly well. However there are some things that need to be clarified before the paper is accepted for publication.

I've also included some suggestions that may improve the final manuscript.

[Figure]

Major comments on the paper and dataset

- Throughout section 4, you provide comments on how the dataset you have created could or should be improved. This is useful, but it also gives the impression that your dataset is not that good after all. It would be better to either a) clarify in the introduction and abstract that this work is simply a first pass, and that more needs to be done, or b) collect all of these comments in a separate section. Perhaps you can include them in section 4.4: limitations and next steps.

- Data: I found it hard to quickly extract information about the datafiles from the figshare website. Can you reproduce Table 1 along with the data?

- Why are you not providing the monthly data through figshare or the doi?

- Many of the data files seem to be relatively binary: black or white. I'm not an expert in rasters so I might be missing something here, but how can I extract the high-resolution detail you are championing in the article?

Minor comments and suggestions

-Page 1, line 4: 'variations' not 'variation'

-Page 3, line 22: are hill fires always human-induced?

-Page 5, line 2: 'temperature buffers' not 'a temperature buffer'

-Page 5, line 22: can you provide a reference to Hong Kong's dense network of stations?

-Page 5, line 27 and 28: I think you should add the word 'absolute' before the variables maximum and minimum temperature, to clarify that these are the highest and lowest temperatures recorded in each month.

-Page 6, line 10: why do you have high confidence in the long-term averaged weather station data?

[Figure]

-Page 7, line 8: please refer to the resolution of the rasters as 30m, to be consistent -

-Page 7, line 15: Aren't you only providing the rasters at one scale?

-Page 8, line 5: Can you provide a brief explanation why the highest maximum temps are in inland valleys?

-Page 8, line 15-16: Aren't you arguing in this study that your new dataset is high resolution? Consider rephrasing this sentence.

-Page 9, line 29: I would say 'our models', rather than 'The new models'

---

## Referee Comment (RC4) · Anonymous Referee #4 · 25 Jan 2019

The authors developed a very high-resolution (30m) gridded dataset of climate, NDVI, and topography for Hong Kong. The meteorological observations from weather stations are interpolated using thin plate spline model. The motivation for fine resolution dataset for Species Distribution Modeling (SDM) is clear and the final product of the study will be useful for SDM and other local applications, however, the manuscript lacks justification for the methodology used and meaningful evaluation of results. It seems to me that the construction of climate data at this high resolution is the novelty of the paper and the main finding (i.e. greater spatial variation in finer resolution data than

the coarser) does not add anything new. The way method section is described is not clear—each variable is prepared separately and then they were used as inputs to the statistical model for the climatology interpolation? Why did the authors choose this method over others and how are the 6 predictors chosen? Also, the use of 'climate modeling' in the text is confusing as it usually refers to general circulation models or regional climate models, but the terminology is used for the spatial interpolation model. I recommend changing the title to something like "development of 30 m raster dataset of climate, vegetation, and topography for Hong Kong" and list specific comments below.

1. Gridded meteorological datasets have been generated using station observations and a variety of interpolation methods in the past. A flagship climate dataset may be the CRU climate data (New et al, 2002) which used thin plate spline technique, with functions of latitude, longitude, and elevation (and mean precipitation for precipitation coefficient of variation). The technique seems to be the standard in recently increasing number of global gridded climatological datasets with increasing spatial resolution (eg. WorldClim2, TerraClim). Additional spatial information that represent physical processes are required in order to resolve higher resolution. I understand a unique situation for Hong Kong for the small domain with dense station network, which may allow simplification compared to constructing global data, but it would be helpful to tie into existing gridded climatology data w.r.t. method of prediction. The paper may shed some lights on improving precipitation interpolation.

2. The stations should be indicated in the map of Hong Kong, Figure 1.

3. Methods: I'm aware that R is a statistical package software. But what is the prediction model used—linear regression? Section 3.2, page 5 line 30- page 6 line 17 describes two-step process, which seems to be the main model (as referred to "our model", "local model", "new model"). Either moving section 3.2 to the first section, or giving an overview of the model before subsections begin, and streamlining the reference to the model will help clarify. Does water proximity include inland water bodies such as river, pond, and wetland? Could NDVI be included as a predictor—wouldn't
it add more physical characteristics? Though annual mean or monthly climatology of NDVI, rather than instantaneous is suitable.

4. As NDVI data is the only remote sensing, physical variable that resolves 30-m, I think it's important to compile climatology. Authors admit that the index values vary seasonally (page 10, line 12), which seems to contradict with the statement earlier on the instantaneous NDVI being representative. With strong seasonality of rainfall pattern in Hong Kong from June to August, I'd expect NDVI would respond. Landsat data extends several decades, so I can't imagine there's not enough data to capture seasonal variation. If no data during monsoon season, dry and winter low and wet summer high would be useful.

5. Precipitation results (4.2.2): I don't understand the last sentence. GCM outputs can be a predictor? If you mean using dynamical models, neither the GCMs nor even higher-resolution regional weather forecast model can't resolve micrometeorlogy at 30-m. Downscaling dynamical model climatology is a possibility but it will be a whole new paper and I'm not sure if it's attainable for 30m with limited information at hand.

6. Climate variables discussion (4.2.4): Though direct validation is not possible, temperature and precipitation could be evaluated qualitatively. Worldclim2 is average for 1970-2000 but your climatology is for 1998-2017, so it's not apple-to-apple comparison. Did you adjust Worldclim data? Could that be the reason for huge discrepancy in precipitation? TerraClimate data set is coarser at $\sim$ 4km but coves 1958-2015 (https://www.nature.com/articles/sdata2017191), so you could get closer climatology of 1998-2015 for the comparison. I would first check if the climatology agrees at station locations, then map out the differences at 4 km. For temperature, you can downscale the 4 km data to 30 m via elevation correction using constant lapse rates of -6.5 °C/km (Willmott and Matsuura, 1995; Maurer et al., 2002) since you have 30 m elevation data which can easily be aggregated up to 4km. The downscaled temperature should provide similar features as the modeled results and physical range of differences to expect. Also, effects of predictors other than elevation would be shown where they differ. Precipitation is difficult to evaluate or even to predict as indicated in the text. Does Hong Kong have radar data?

7. Skin temperature from Landsat could be another data to evaluate the heterogeneity of the modeled temperature. Though skin temperature is not exactly the same as in-situ 2-m air temperature, it is an observation based, independent data.

8. My understanding is that bilinear interpolation is for coarser to finer spatial interpolation and for aggregating from finer to coarser, arithmetic or area weighted average is appropriate. I'm wondering if using bilinear to aggregate from 30m to 1 km (Figure 7 etc.) results in different 1km if arithmetic averaging is used.

9. Next step: It is important to note what's missing and limited for future enhancement, but you should also encourage people to use this dataset. Isn't the dataset ready to use in SDM to address the issues raised in the introduction section? 30 m is remarkably high resolution and the entire raster data contain valuable information for many modeling studies and local management applications.

---

## Author Comment (AC1) · 26 Mar 2019

Dear Anonymous Referee 1, Thank you very much for reviewing the manuscript and providing your feedback and concerns. Below we provide point to point responses (AC) to your comments (RC), as well as changes in the manuscript (CM). Page and line numbers refer to those in the submitted manuscript. We also provide an attached pdf document showing tracked changes, new citations, figures, and an appendix added to the original manuscript.

[Figure]

On behalf of the authors,
Brett Morgan

**RC** - Referee comment     **AC** - Author comment     **CM** - Change in the manuscript

**RC1.01** The manuscript is not sufficiently organized and confused with no novelty and explicit research question. There are many too short subsections, which should be merged. Methods are not much clear because details and relevant references have not been provided. Consequently, it is not much easy to follow results and discussion.
**AC1.01** We agree that improving the clarity and organization of the manuscript is necessary, though challenging because of the large number of data inputs, outputs, and analyses. We have restructured and added to sections (especially Section 3 - Methods and Section 4 - Results and Discussion) to improve clarity. The novelty of the manuscript is the data itself, as stated on Page 1, Line 9: "To our knowledge, this is the first set of published environmental rasters specific to Hong Kong."; Page 4, Lines 1-3: "Therefore Hong Kong is in dire need of a comprehensive suite of accessible environmental GIS data, at a resolution finer than 1 km, suitable for species distribution modeling and other local applications. To this end, we developed new, 30 m resolution rasters of topography, NDVI, and interpolated climate variables for each month of the year."; and Page 10, Line 25: "This diverse set of 30 m resolution topography, climate, and remote sensing data include the first published interpolation of long-term climate averages specific to Hong Kong." Please see AC1.06 for our response regarding a research question. As most readers will likely use only parts of the provided data, we believe that retaining the subsections will help the reader quickly find information of relevance for the data they want to use. Lumping subsections together would likely add to the confusion mentioned.

**RC1.02** The Authors have used data associated at support sizes very different. They should take into account the change of support.
**AC1.02** We are uncertain what the reviewer means by "data associated at support sizes very different," and would appreciate further explanation. If the concern is that

input rasters used as model predictors were initially at different resolutions, higher resolution products were resampled to 30 m before model building (Page 4, Lines 15-16).

**RC1.03** The title should be made more informative and effective.
**AC1.03** We have reformulated the title to make it more informative and better reflect the focus of the manuscript. We welcome additional suggestions on how it could be improved.
**CM1.03** Title: New 30 m resolution Hong Kong climate, vegetation, and topography rasters indicate greater spatial variation than global grids within an urban mosaic

**RC1.04** The Abstract has not the required structure and does not summarize the whole manuscript. It should be organized better and explain clearly what was done, what was found and what are the main conclusions. Generally, the first sentence should provide briefly the rational of the topic being investigated.
**AC1.04** We are not aware of abstract structure requirements that this abstract does not adhere to. In the ESSD manuscript preparation guidelines for authors, it is stated "The abstract should be intelligible to the general reader without reference to the text. After a brief introduction of the topic, the summary recapitulates the key points of the article and mentions possible directions for prospective research. Reference citations should not be included in this section, unless urgently required, and abbreviations should not be included without explanations. Please include the DOI(s) to the referenced data set(s) as well as the citation(s)."

**RC1.05** Keywords are missing.
**AC1.05** We would happily provide keywords, but we did not find a format for them in the Earth Systems Science Data LaTeX template, and published papers in ESSD do not have keywords.

**RC1.06** The Introduction section is confused ant not sufficiently organized. Particularly, reading the title, one is expecting to find in the Introduction the presentation of what the title promises, but unfortunately it is not so. The Introduction should be improved and

the topic being investigated should be explained clearly. The novelty and objectives are missing. A manuscript to be considered a research paper, a research question must be clearly stated. In addition, the Authors should explain the gap in the topic being investigated and how their study fills such a gap.

**AC1.06** We hope the changes in the title resolve the stated discrepancy in the introduction. Many of the missing elements (novelty, objectives, research gap, research question) are present in section 2 about the study area, which is meant to be an extension of the introduction. For example, the knowledge gap is that Hong Kong is lacking appropriate resolution data for local applications (Page 4, Line 1). The order of these elements could be rearranged, but it seems less logical to pose this research question and the objective of developing higher resolution rasters before introducing Hong Kong and the existing GIS data available for it. Alternatively, sections 1 and 2 (Introduction and Study Area) could be merged into a single large introduction section. However we believe keeping these sections separate allows the reader to more easily navigate to content of interest. We are skeptical that a central research question is necessary for this manuscript. Much scientific research is indeed hypothesis-driven, but in alignment with the title of this journal, Earth System Science Data, our project is data-driven. In the "About" section of the ESSD website, it is stated "Articles in the data section may pertain to the planning, instrumentation, and execution of experiments or collection of data. Any interpretation of data is outside the scope of regular articles." In agreement with this defined scope, our primary goal in writing this manuscript is to describe the development of the provided data, rather than answering a central question.

**RC1.07** A well-organized Materials and Methods section is missing. The sections '2 Study area' and '3 Methods' should be included in a new Materials and Methods section which allows readers to follow the progress of the objectives in the manuscript and support results and discussion. In the methods, how data have been analysed and combined should be explained providing sufficient details. Particularly, the Authors should explain how they have taken into account the change of support problem to have all data associated to the same support size. Details and references on statistical

methods are missing.

**AC1.07** We share your concerns on the methods section, which we have improved with various changes in structure, additional statistical details, and references throughout. Specifically we have better explained the meaning of each variable and the reasoning behind their development. We do not believe that merging the methods section with section 2, "Study Area," would be beneficial. Section 2 is largely descriptive and doesn't cover any of the materials (data sources) used in the analyses, so the content would be out of place in a Materials and Methods section. As said in AC1.02, we are unsure what is meant by support size, and we would appreciate further explanation.

**RC1.08** Results and Discussion sections should be improved and supported by a new Materials and Methods section.

**AC1.08** For the Materials and Methods sections, please refer to AC1.07. The results and discussion section has been modified to improve the clarity and content of the manuscript. This has included creation of section 4.5 "Limitations and next steps" and section 4.4 "Value and Utility," which discusses the results in consideration of how they will enable SDM and other environmental research in this important region.

**RC1.09** Conclusions are poor: they should be improved and to show the improvement of our knowledge.

**AC1.09** Thank you for this feedback, we agree that improved conclusions are desirable. We believe the improvement in our knowledge is summarized in the first sentence of the conclusions: "This diverse set of 30 m resolution topography, climate, and remote sensing data include the first published interpolation of long-term climate averages specific to Hong Kong."

Please also note the supplement to this comment:
https://www.earth-syst-sci-data-discuss.net/essd-2018-132/essd-2018-132-AC1-supplement.pdf

[Figure]

**Supplement:**

[revised manuscript text omitted]

between 100 m and 400 m elevation (Fig. 8), and is distributed between Hong Kong Island, Lantau Island, and the New Territories. One exception is the verdant mangrove forests, at sea level. The patchy distribution of high density vegetation likely reflects the effects of historical deforestation. The largest patches are found on the southeastern slopes of Tai To Yan in the New Territories. The relative distribution of NDVI classes along Hong Kong's elevational gradient is shown in Figure 8, Future work could determine to what extent NDVI changes over time, in response to seasonality or recent weather. The limiting factor is the availability of data of adequate temporal resolution, as many satellite images of Hong Kong are obscured by cloud cover or degraded by poor air quality.

**4.4 Value and Utility**

This new data will benefit environmental research, and specifically SDM studies, in two main ways. First, it will enable finer scale analyses than previously possible. For SDM, this means improved detection of climatic microrefugia (Meineri and Hylander, 2017), and the ability to differentiate between human altered habitat and natural areas. Rampant development and a shifting climate make this knowledge of local species persistence more important than ever. Additionally, this is especially relevant in Hong Kong where topography varies dramatically, and where urban areas form a complex mosaic with undeveloped expanses.

Second, we provide a diverse array rasters derived from multiple independent data sources, but in a single resolution and format to facilitate further analysis and synthesis of meaning. For SDM, these diverse layers have distinct advantages over datasets that only contain climate data. Compared to climate data alone, using diverse predictors including topographic characteristics have been shown to be important variables for accurate SDM results, such as predicting the spread of invasive species in new ranges (Peterson and Nakazasa, 2008). However benefits of non-climate data may only be evident in finer scale SDMs (Luoto et al., 2007).

Finally, such high quality, diverse geographic data is especially uncommon in tropical regions, where improved knowledge for environmental research and biological conservation is most needed. According to Rapoport's Rule, tropical species are more likely to have smaller distributions (Stevens, 1989), and therefore future execution of local SDM studies to understand their ranges are particularly important.

**4.5 Limitations and next steps**

Here we outline how shortfalls of the data presented may be improved in the future. First, though we inferred Hong Kong's pattern of urban development from impervious surface data, this is less than ideal because in addition to concrete, bare soil or rock are sensed as impervious. Also, it cannot differentiate dense urban cores of high-rises from large paved areas. For climate modeling, an urbanicity measure that considers building height or population density at a 30 m or finer scale could be preferable.

Second, while our temperature rasters should accurately represent air temperature in open areas, they do not reflect the high spatial variation in temperature found in urban microclimates. For

[revised manuscript text omitted]

Moved (insertion) [4]

Figures and Appendix Added:

---

## Author Comment (AC2) · 26 Mar 2019

Dear Anonymous Referee 2,
Thank you very much for reviewing the manuscript and providing your feedback. Below we provide point to point responses (AC) to your comments (RC), as well as changes in the manuscript (CM). Page and line numbers refer to those in the submitted manuscript. We also provide a pdf supplement showing tracked changes, new citations, figures, and an appendix added to the original manuscript.

[Figure]

On behalf of the authors,
Brett Morgan

**RC** - Referee comment    **AC** - Author comment    **CM** - Change in the manuscript

**RC2.01** This study aims to produce a high resolution (30 m raster) data set of climate and environmental variables for the Hong Kong region. Unfortunately, I find the manuscript to be confusing, showing an overall disconnection between sections. The manuscript focuses on a large but incomplete description of the variables included in the data set, and does not address the main conclusion stated in its title ("Local models reveal greater spatial variations than global grid in an urban mosaic").

**AC2.01** Changes addressing these concerns have been made throughout the manuscript. In our modifications, we attempted to make the manuscript more cohesive, with more explanation of how the various data are related. This included completion of the methods section with more detailed descriptions of variables. We have added Figure S1, a flow chart which we hope illustrates connections between sections as well as the data provided. We have altered the title to make it more informative and better reflect the focus of the manuscript.

**CM2.01** Title: New 30 m resolution Hong Kong climate, vegetation, and topography rasters indicate greater spatial variation than global grids within an urban mosaic

**RC2.02** The introduction section discusses the application of "Species distribution modeling (SDM)" and how this type of analysis is affected by the spatial resolution of the environmental data employed. However, this introductory discussion seems to be irrelevant within the context of the manuscript, as SDM is rarely mentioned again throughout the text. Abbreviations such as NDVI are used throughout the abstract and introduction but are not explained until the later sections of the methods section.

**AC2.02** We agree that the omission of meaningful discussion of SDM implications was an oversight. We have added this in Section 4.4 Value and Utility, including how the results will benefit SDM studies and why this improvement in our knowledge is much

needed. We have clarified the meaning of NDVI in the abstract and introduction.

**CM2.02** Page 1, Line 7: The data include topographic variables, Normalized Difference Vegetation Index, and interpolated climate variables based on weather station observations.

Page 2, Line 21: For example, vegetation measures like the Normalized Difference Vegetation Index (NDVI) in fragmented forests are un-likely to be relevant if the grain size is much larger than the forest patch size, because each grid cell will be a single averaged value.

**RC2.03** In the method section each of the topographic and climate variables, as well as remote sensing products are mentioned. However, it seems to me that each of the subsections focuses on irrelevant details, and there is no clear descriptive explanation of a) what these variables are? b) why were they chosen? and c) how were they processed?

**AC2.03** We have modified the various methods subsections where this information was missing, adding more description of the meaning of each variable and stating that variables were chosen based on the availability of source data, as well as their expected utility in SDM research. We have added Appendix 1, which provides defini-tions of all climate and topography variables for easy reference. We have also added Figure S1, which shows the general raster workflow, helping explain how variables were processed.

**AC2.03** Page 4, Line 8: The variables developed were selected based on their utility in environmental research, especially SDM, as well as the availability of appropriate source data.

**RC2.04** The results and discussion section is also vague and difficult to read. There is no clear distinction between the validation data set/model and the novel data/model analysis produced by this study. The figures lack explanation within the main text, and it is hard to see how they convey the results of the study.

**AC2.04** In the results and discussion section, we have reorganized much of the

content including merging research limitations into one section, and adding discussion of the results in light of potential SDM applications. We added sentences at the end of the paragraph describing cross-validation procedures, to clear up the distinction between the validation and final rasters produced. We have added additional figure references in the text, and we would welcome suggestions on how the figures could better convey the results.

**CM2.04** Page 6, Line 17: This cross-validation procedure was used only to produce these validation measurements. The finalized monthly climate rasters described above were trained using all available data.

**RC2.05** Overall, I believe this manuscript needs substantial revisions, and perhaps a reassessment of the scientific goals that it is trying to communicate.

**AC2.05** We believe we have fixed the main problem, which was that the previous manuscript title was misleading regarding the main scientific goals. Other sections of the manuscript have been modified to reflect this, and more detail and explanation has been added to improve clarity.

Please also note the supplement to this comment:
https://www.earth-syst-sci-data-discuss.net/essd-2018-132/essd-2018-132-AC2-supplement.pdf

[Figure]

**Supplement:**

[revised manuscript text omitted]

---

## Author Comment (AC3) · 26 Mar 2019

Dear Anonymous Referee 3,
Thank you very much for reviewing the manuscript and providing your feedback. Below we provide point to point responses (AC) to your comments (RC), as well as changes in the manuscript (CM). Page and line numbers refer to those in the submitted manuscript. We also provide a pdf supplement showing tracked changes, new citations, figures, and an appendix added to the original manuscript.

[Figure]

On behalf of the authors,
Brett Morgan

**RC** - Referee comment    **AC** - Author comment    **CM** - Change in the manuscript

**RC3.01** Throughout section 4, you provide comments on how the dataset you have created could or should be improved. This is useful, but it also gives the impression that your dataset is not that good after all. It would be better to either a) clarify in the introduction and abstract that this work is simply a first pass, and that more needs to be done, or b) collect all of these comments in a separate section. Perhaps you can include them in section 4.4: limitations and next steps.
**AC3.01** We have combined and condensed the discussion of data limitations into Section 4.4: Limitations and next steps. Thank you for the suggestion.

**RC3.02** Data: I found it hard to quickly extract information about the datafiles from the figshare website. Can you reproduce Table 1 along with the data?
**AC3.02** Thank you for this suggestion, we have added a file in the figshare repository equivalent to Table 1 of the manuscript, showing file descriptions, units, and raster summary statistics.

**RC3.03** Why are you not providing the monthly data through figshare or the doi?
**AC3.03** The thought was that having another 120 raster files in the repository would complicate finding and downloading the desired files, especially because we expect only the yearly summary layers would be of interest to most users. We decided to compress the monthly models into a single zip file now available in the repository. This way, they are available for those who are interested but avoid user confusion.
**CM3.03** Page 11, Line 4: Individual monthly rasters for each of the 10 climate variables are available as a compressed zip file.

**RC3.04** Many of the data files seem to be relatively binary: black or white. I'm not an expert in rasters so I might be missing something here, but how can I extract the
high-resolution detail you are championing in the article?

**AC3.04** If we understand correctly, the black and white you describe is referring to the file previews shown on the figshare website. Unfortunately figshare seems to have this standard rendering of raster files, displaying the preview as a binary image. To access the raster data directly and display it how you want, you would need to download the files and open them in GIS software, such as QGIS.

**RC3.05** Page 1, line 4: 'variations' not 'variation'

**AC3.05** We adopted this suggestion.

**CM3.05** Further, these global datasets likely underestimate local climate variations because they do not incorporate locally relevant variables.

**RC3.06** Page 3, line 22: are hill fires always human-induced?

**AC3.06** As far as is known, yes. In Hong Kong, lightning only occurs during heavy rain, usually during the monsoonal summer. The vast majority of these fires happen during the dry winter, with spikes in frequency associated with holidays and religious practices where burning in hillside cemeteries is practiced. (see Chau, 1994: http://hub.hku.hk/handle/10722/34430)

**RC3.07** Page 5, line 2: 'temperature buffers' not 'a temperature buffer'

**AC3.07** We adopted this suggestion.

**CM3.07** Water bodies adjacent to land areas can act as temperature buffers, contribute to evaporative cooling (Lookingbill and Urban, 2003), and influence precipitation patterns (Heiblum et al., 2011; Paiva et al., 2011); therefore considering their distribution is important for climatic predictions.

**RC3.08** Page 5, line 22: can you provide a reference to Hong Kong's dense network of stations?

**AC3.08** Yes, we have added a reference.

**CM3.08** In contrast, interpolation in Hong Kong is benefitted by a relatively small geographic area and a quite dense network of weather data provided by dozens of perma-

nent weather stations (Hong Kong Observatory, 2018).

**RC3.09** Page 5, line 27 and 28: I think you should add the word 'absolute' before the variables maximum and minimum temperature, to clarify that these are the highest and lowest temperatures recorded in each month.

**AC3.09** We did consider adding the word "absolute" to these variables, but this might add to confusion about the meaning of the measurement. "Absolute maximum temperature" of a given month might normally refer to the highest temperature ever recorded in that month, but what we provide instead is data that represents the averaged absolute maximum values recorded over a period of 20 years. To ensure clarity of the meaning of each variable, we have added Appendix 1, which provides definitions for easy reference.

**RC3.10** Page 6, line 10: why do you have high confidence in the long-term averaged weather station data?

**AC3.10** The confidence is based on the good availability of measurements for averaging - a weather station's data was only included if at least 8 full years of measurements were available for use, and most stations had many more years than that threshold. So we can be fairly confident that the averages are good approximations of the true climate at each station.

**CM3.10** This low lambda value was selected because of the relatively high confidence in the long-term averaged weather station values (based on at least 8 years of data).

**RC3.11** Page 7, line 8: please refer to the resolution of the rasters as 30m, to be consistent.

**AC3.11** We adopted this suggestion.

**CM3.11** All rasters are provided at an identical 1 arc second (30 m) resolution and in the WGS84 geographic coordinate system.

**RC3.12** Page 7, line 15: Aren't you only providing the rasters at one scale?

**AC3.12** The rasters are all at one resolution (30 m), but the values of these topographic

variables were calculated using buffers at multiple scales. For example, relative eleva-
tion at a given 30 m pixel will vary depending on the size of the surrounding area (in
our case, a circle of a given radius) to which it is compared. We adjusted the wording
to help clarify.

**CM3.12** For this reason, we provide these rasters calculated at multiple buffer scales.

**RC3.13** Page 8, line 5: Can you provide a brief explanation why the highest maximum
temps are in inland valleys?

**AC3.13** Yes, and we have also added a sentence about how the difference in maximum
vs. minimum temperature patterns can be explained by urban heat island effects.

**CM3.13**  This pattern may be explained by urban heat retention: buildings act as heat
sinks which absorb solar radiation during the day, and slowly release heat at night,
causing increased minimum temperatures. The high maximum temperatures in inland
valleys may be due to reduced air circulation in sheltered locations, and lack of complex
vegetation or urban structures providing shade.

**RC3.14** Page 8, line 15-16: Aren't you arguing in this study that your new dataset is
high resolution? Consider rephrasing this sentence.

**AC3.14** Not exactly. We do want to highlight that our rasters have a much higher res-
olution than similar datasets that are available globally. However, describing a raster
simply as "high" or "low" resolution is quite arbitrary, as many 1 km datasets are de-
scribed as "high resolution." For certain applications, 30 m resolution would be quite
low. Assessment of urban microclimate is one of those cases, and we attempt to con-
vey this.

**RC3.15** Page 9, line 29: I would say 'our models', rather than 'The new models'
**AC3.15** We adopted this suggestion.
**CM3.15** Our models generally indicate greater spatial variation than Worldclim, with
cool areas colder, warm areas hotter, and wet areas wetter.

Please also note the supplement to this comment:

https://www.earth-syst-sci-data-discuss.net/essd-2018-132/essd-2018-132-AC3-supplement.pdf

[Figure]

**Supplement:**

[revised manuscript text omitted]

---

## Author Comment (AC4) · 26 Mar 2019

Dear Anonymous Referee 4,
Thank you very much for reviewing the manuscript and providing your feedback. Below we provide point to point responses (AC) to your comments (RC), as well as changes in the manuscript (CM). Page and line numbers refer to those in the submitted manuscript. We also provide a pdf supplement showing tracked changes, new citations, figures, and an appendix added to the original manuscript. This comment appears to have had some text encoding errors which we have left intact.

[Figure]

On behalf of the authors,
Brett Morgan

**RC** - Referee comment    **AC** - Author comment    **CM** - Change in the manuscript

**RC4.01** The authors developed a very high-resolution (30m) gridded dataset of climate, NDVI, and topography for Hong Kong. The meteorological observations from weather stations are interpolated using thin plate spline model. The motivation for fine resolution dataset for Species Distribution Modeling (SDM) is clear and the final product of the study will be useful for SDM and other local applications, however, the manuscript lacks justification for the methodology used and meaningful evaluation of results. It seems to me that the construction of climate data at this high resolution is the novelty of the paper and the main finding (i.e. greater spatial variation in finer resolution data than the coarser) does not add anything new. The way method section is described is not clear; each variable is prepared separately and then they were used as inputs to the statistical model for the climatology interpolation? Why did the authors choose this method over others and how are the 6 predictors chosen? Also, the use of 'climate modeling' in the text is confusing as it usually refers to general circulation models or regional climate models, but the terminology is used for the spatial interpolation model. I recommend changing the title to something like "development of 30 m raster dataset of climate, vegetation, and topography for Hong Kong" and list specific comments below.

**AC4.01** We agree that the main finding and novelty of this study is the higher resolution of the developed rasters. To reflect this focus and better represent the contents of the manuscript, we have modified the title as seen below. However, we believe that the findings of greater spatial variation in climate results is still a salient component of the study. Although this is a result that might be expected, it will have important conse-quences in projects that use this data, in particular for species distribution modelling for which changes in a few degrees can substantially modify final outcomes. Users should know that not only is the resolution different from products like WorldClim, but also that
the values are different. We also believe the increased variation indicates that global climate interpolation data exclude climate forcing factors that are relevant at smaller scales, which is an important result. We have made various changes and clarifications in the methods section, and provide a workflow schematic in Figure S1 to illustrate preparation of all of the rasters and variables. The thin plate spline methodology was used because of the availability of tools in the R environment to implement it, as well as its history of use in climate interpolation research, which we have now addressed in the manuscript. To select the six climate predictors, we searched the literature for what types of variables have been used in similar studies in the past, and then used those that we expected to have climatic effects at the geographic extent and scale of this study.

**CM4.01** Title: New 30 m resolution Hong Kong climate, vegetation, and topography rasters indicate greater spatial variation than global grids within an urban mosaic
Page 5, Line 31: Independent variables were selected by searching the literature for similar studies, and choosing predictors we expected to have an influence on climate at this regional scale.

**RC4.02** Gridded meteorological datasets have been generated using station observations and a variety of interpolation methods in the past. A flagship climate dataset may be the CRU climate data (New et al, 2002) which used thin plate spline technique, with functions of latitude, longitude, and elevation (and mean precipitation for precipitation coefficient of variation). The technique seems to be the standard in recently increasing number of global gridded climatological datasets with increasing spatial resolution (eg. WorldClim2, TerraClim). Additional spatial information that represent physical processes are required in order to resolve higher resolution. I understand a unique situation for Hong Kong for the small domain with dense station network, which may allow simplification compared to constructing global data, but it would be helpful to tie into existing gridded climatology data w.r.t. method of prediction. The paper may shed some lights on improving precipitation interpolation.

**AC4.02** Thank you! We agree it is important to make clear that this methodology has

been used much in the past as well as recently in climate interpolation studies. We have added explanation in the methods to reflect this.

**CM4.02** Page 5, Line 22: Here we use multiple linear regression to predict geographic climate patterns using weather station training points and raster covariates. This is followed by thin plate spline (TPS) interpolation (see Wahba, 1979) of the regression model residuals. TPS is a widely used approach in climate interpolation (e.g. New et al., 2002; Fick and Hijmans, 2017), which fits a curved surface to irregularly distributed points. This two-step interpolation (regression followed by TPS) was based on the approach of Meineri and Hylander (2017).

**RC4.03** The stations should be indicated in the map of Hong Kong, Figure 1.

**AC4.03** Adding the stations to Figure 1 would make it quite busy due to the density of stations in Hong Kong, so instead we plotted a new map as a supplemental figure showing the distribution of weather stations. It shows stations from which temperature or rainfall measurements are available. It is also used an opportunity to display elevation more clearly.

**CM4.03** Added Figure S2.

**RC4.04** Methods: I'm aware that R is a statistical package software. But what is the prediction model used ÌȨTĚĞlinear regression? Section 3.2, page 5 line 30- page 6 line 17 describes two-step process, which seems to be the main model (as referred to "our model", "local model", "new model"). Either moving section 3.2 to the first section, or giving an overview of the model before subsections begin, and streamlining the reference to the model will help clarify. Does water proximity include inland water bodies such as river, pond, and wetland? Could NDVI be included as a predictor ÌȨTĚĞwouldn't it add more physical characteristics? Though annual mean or monthly climatology of NDVI, rather than instantaneous is suitable.

**AC4.04** We have added an overview section as suggested (shown in CM4.02) that makes clear the modeling method is linear regression, with other relevant explanation. Water proximity does consider inland water bodies, most of which are artificial reservoirs. NDVI could be used as a predictor, but it is variable on a granular scale where neighboring pixels can have very different values. It is unlikely that climate variables would vary in this way, so we decided not to include NDVI as a predictor. Also, even if they are surrounded by vegetation, most weather stations are likely positioned on some type of structure that would bias local NDVI measurements.

**CM4.04** Page 5, Line 5: Second, water proximity (including inland water bodies) was calculated as the percent surface land in the area surrounding a given pixel.

**RC4.05** As NDVI data is the only remote sensing, physical variable that resolves 30-m, I think it's important to compile climatology. Authors admit that the index values vary seasonally (page 10, line 12), which seems to contradict with the statement earlier on the instantaneous NDVI being representative. With strong seasonality of rainfall pattern in Hong Kong from June to August, I'd expect NDVI would respond. Landsat data extends several decades, so I can't imagine there's not enough data to capture seasonal variation. If no data during monsoon season, dry and winter low and wet summer high would be useful.

**AC4.05** We are unsure what is meant by "compile climatology," in relation to NDVI, and if our following response does not address this comment we would appreciate further explanation. We agree that discussion of the NDVI data may appear contradictory, and would like to emphasize that while the NDVI values may fluctuate, we expect that the overall geographic pattern of NDVI (highest in dense forests, lowest in urban centers) remains fairly consistent throughout the year. While precipitation is indeed strongly seasonal, the vegetation in Hong Kong is not seasonally deciduous, but evergreen, so changes in NDVI are unlikely to be drastic. Exceptions may include agricultural areas with rapid shifts associated with harvest cycles. The difficulty in acquiring suitable Landsat images for Hong Kong stems from several factors, as touched on within the manuscript. First, cloud cover is a hindrance, and overcast skies are especially common in the first half of the year. For the months of June-August, mean cloud cover (at the Hong Kong Airport weather station) can range from 65 to 80 percent. So

most Landsat images captured during this time are entirely obscured, and very few are cloud-free. Second, no Landsat image covers the entirety of Hong Kong: it is on the edge of the satellite path, and so generating a complete NDVI raster requires finding and merging two or more suitable images taken at a similar time of the year.

Upon checking Landsat 8 further, creating a dry season / winter NDVI layer would certainly be possible, but for calculating summer NDVI not enough cloud-free images are available. We will check Landsat 7, 5, and 4 databases as they are also available at 30 m resolution, but a concern with using older data that is that in comparisons some NDVI differences will be due to succession of vegetation and disturbance of lands due to continuous development over the years rather than seasonality.

**RC4.06** Precipitation results (4.2.2): I don't understand the last sentence. GCM outputs can be a predictor? If you mean using dynamical models, neither the GCMs nor even higher-resolution regional weather forecast model can't resolve micrometeorlogy at 30-m. Downscaling dynamical model climatology is a possibility but it will be a whole new paper and I'm not sure if it's attainable for 30m with limited information at hand.

**AC4.06** Thank you, yes it makes sense that circulation models wouldn't be helpful without first downscaling them, and indeed that seems like it would be a substantial effort outside the scope of this study.

**CM4.06** The sentence has been removed.

**RC4.07** Climate variables discussion (4.2.4): Though direct validation is not possible, temperature and precipitation could be evaluated qualitatively. Worldclim2 is average for 1970-2000 but your climatology is for 1998-2017, so it's not apple-to-apple comparison. Did you adjust Worldclim data? Could that be the reason for huge discrepancy in precipitation? TerraClimate data set is coarser at âĹij 4km but coves 1958-2015 (https://www.nature.com/articles/sdata2017191), so you could get closer climatology of 1998-2015 for the comparison. I would first check if the climatology agrees at station locations, then map out the differences at 4 km. For temperature, you can downscale the 4 km data to 30 m via elevation correction using constant lapse rates of

[Figure]

-6.5◦C/km (Willmott and Matsuura, 1995; Maurer et al., 2002) since you have 30 m elevation data which can easily be aggregated up to 4km. The downscaled temperature should provide similar features as the modeled results and physical range of differences to expect. Also, effects of predictors other than elevation would be shown where they differ. Precipitation is difficult to evaluate or even to predict as indicated in the text. Does Hong Kong have radar data?

**AC4.07** In our comparisons with WorldClim, the primary goal was to assess the geographic differences between the two datasets. So if in choosing a dataset for comparison, there is a tradeoff between higher resolution rasters (WorldClim 2) and a more congruent temporal window (TerraClim), we prefer to use the dataset with higher resolution. We did not adjust the WorldClim values. We did consider testing model predictions (both our models and WorldClim) against actual station values, but it seems there would be an issue of testing with data that was used to train one model but not the other. As for downscaling 4 km TerraClimate data down to 30 m based on elevation only, we are unsure that would allow for a valid comparison. Temperature lapse rates would likely vary by geographic region, as well as the temperature variable under consideration. The Hong Kong Observatory does record radar data (shown here https://bit.ly/2FyS4Rj), but it doesn't seem historical radar data are available for download.

**RC4.08** Skin temperature from Landsat could be another data to evaluate the heterogeneity of the modeled temperature. Though skin temperature is not exactly the same as in-situ 2-m air temperature, it is an observation based, independent data.

**AC4.08** Thank you for this interesting proposition. Is skin temperature the same as thermal infrared images (Landsat provides two thermal infrared bands, but only at 100 m resolution)? It seems this could be quite biased by many factors, like the ground cover, sun intensity at the moment the image was taken, etc. While we were able to put together NDVI, we are no remote sensing experts, and so would need some more guidance on how validation using Landsat data might be accomplished.

[Figure]

**RC4.09** My understanding is that bilinear interpolation is for coarser to finer spatial interpolation and for aggregating from finer to coarser, arithmetic or area weighted average is appropriate. I'm wondering if using bilinear to aggregate from 30m to 1 km (Figure 7 etc.) results in different 1km if arithmetic averaging is used.

**AC4.09** We are not familiar with using arithmetic or area weighted average interpolation, but our understanding is that bilinear interpolation is generally appropriate to use for resampling continuous rasters. We think the different methods would likely produce different results, but then the challenge would be to evaluate which is actually superior to the other in terms of accuracy.

**RC4.10** Next step: It is important to note what's missing and limited for future enhancement, but you should also encourage people to use this dataset. Isn't the dataset ready to use in SDM to address the issues raised in the introduction section? 30 m is remarkably high resolution and the entire raster data contain valuable information for many modeling studies and local management applications.

**AC4.10** Thank you! We have consolidated limitations into one section (4.4), and have tried to be more positive about the opportunities offered by the newly developed rasters in our discussion.

Please also note the supplement to this comment:
https://www.earth-syst-sci-data-discuss.net/essd-2018-132/essd-2018-132-AC4-
supplement.pdf

---

## Author Response (AR2)

[revised manuscript text omitted]

Response to Anonymous Referee #5

Dear Anonymous Referee #5,
Thank you very much for reviewing the manuscript and providing your feedback and concerns. Below we provide point to point responses (AC) to your comments (RC), as well as changes in the manuscript (CM). Page and line numbers refer to those in the first manuscript revision. We also provide an attached pdf document showing tracked changes.

On behalf of the authors,
Brett Morgan

**RC** - Referee comment          **AC** - Author comment          **CM** - Change in the manuscript

**RC5.01** This study presents a set of interpolated bioclimatic variables (plus other variables) specific for Hong Kong, a locale marked by dramatic ecological gradients and interanual climatic variation. The novelty of the study lies in its regional specificity and use of high-resolution DEM and satellite data to make predictions on a 30 m scale. Furthermore, the study is well executed and well written. Somewhat unsurprisingly the high-resolution dataset for this locale was different from the global dataset, with greater spatial variability and more intense extremes. Unfortunately it was not actually demonstrated that the higher resolution data produced improved species distribution models, although the introduction seemed to be setting this up and this was implied throughout (I sense a companion piece may be forthcoming). Furthermore the first half of the results/discussion section is essentially a summary of the climate and topography of HK -- OK, but what new information do the rasters produce compared to climate station summaries? The overall result is that the sections feel a little disjointed, but that may certainly be acceptable for a data paper in this journal.
**AC5.01** You are correct that a companion piece is forthcoming! Here we wanted to make sure all this data is freely available for others to use as well.

Major Points:

**RC5.02** Errors for each climate variable in units of observation (degrees, mm, etc) should be presented in a table. This would allow for some level of comparison with other datasets (although authors note how this is not truly possible). For instance, I would expect that overall error in temperature (which is fairly easy to predict) would not differ much from the global dataset (a couple degrees), but other variables such as precipitation might be more accurate.
**AC5.02** In hindsight we agree that such unadjusted error measurements would be useful for comparing to other datasets, but unfortunately we did not retain that information during the modeling process.

**RC5.03** The authors should at least consider the critique offered by Hijmans (2012) and others about the use of randomly-sampled cross-validation groups in spatial applications. Namely, control and validation points located near each other by chance may artificially reduce global error in a way that is not reflective of model performance. It wasn't clear if validation groups were spatially stratified, but this should be mentioned.
Hijmans, R. J. Cross-validation of species distribution models: removing spatial sorting bias and calibration with a null model. Ecology 93, 679–688 (2012).
**AC5.03** While Hijmans (2012) discussed this issue in the context of SDMs where binary and presence absence points are used to train models, ours are based on measurements of continuous variables and have no 'absence' points. Therefore we are uncertain the concept of spatial sorting bias as described in the Hijmans paper is still applicable here. Regardless, we agree that it will be good to mention.

**CM5.03** Section 3.2 - While randomly selected test points may be subject to spatial sampling bias (Hijmans 2012), this may be less of a concern for this study because in Hong Kong the weather stations are fairly stratified (Figure S2).

Minor Points:

**RC5.04** abstract (line 3) -- As written, sounds as if the authors are saying that 1km data are too coarse for SDM, which they are not. They are too coarse for precision applications and certain contexts.
**AC5.04** Because it could be interpreted in that way, we have added "regional" to describe the SDMs.
**CM5.04** However these data, often 1 km at the finest scale available, are too coarse for applications such as precise designation of conservation priority areas and regional species distribution modeling, or purposes outside of biology such as city planning and precision agriculture.

**RC5.05** page2 paragraph 1 -- a general SDM/review paper citation here would be helpful
**AC5.05** The majority of that paragraph (sentences 1-4) can be attributed to the subsequent citation (Peterson et al., 2011), a book which broadly covers many aspects of SDM theory, methods, and applications. We have added a second citation for it at the end of sentence two.

**RC5.06** Page3 line 7 -- Careful! Many 'flat deserts' have arguably more ecologically important variation at small scales, related to soil properties. I would generalize this to something like 'contexts with more gradual environmental transitions'
**AC5.06** We think it is helpful to have concrete examples of habitats here, so instead of saying deserts likely vary less we changed it to say they "may" vary less.
**CM5.06** Lastly, the utility of fine grain environmental grids can depend on habitat; flat deserts may have less biologically relevant fine-scale spatial variation compared to mountainous forests or subtropical areas fragmented by human activity, like Hong Kong.

**RC5.07** Page 4 line 29 -- Just a thought -- I'd be curious how a 5m resolution model would compare to the 30m resolution, and if the pattern of underestimated climatic variation would continue at this scale.
**AC5.07** We are curious too! We expect it would continue but at a smaller magnitude. However compared to the 30 m models, raster calculations on 5 m data would in theory take 36x as long. Also, at that resolution we might start running into uncertainty related to the training points: imprecision of weather station GPS coordinates and the physical placement of different instruments at stations could more easily cause placement in an incorrect grid cell.

**RC5.08** page 5 line 18 -- cite raster package
**CM5.08** Hijmans, R. J.: Package 'raster'. Geographic Data Analysis and Modeling. R package version 2.8-19. 2019.

**RC5.09** page 5 line 27 -- how was collinearity tested?
**AC5.09** We have added some details.
**CM5.09** The six model predictors were tested for collinearity using vifstep() in the *usdm* R package (Naimi et al., 2014) with a variance inflation factor threshold of 6, and no problems were found.

**RC5.10** page 6 line 33 -- cite fields package
**CM5.10** Nychka, D., Furrer, R., Paige, J., and Sain, S. Package 'fields' Tools for spatial data. R package version 9.6. 2017.

**RC5.11** Page 7 line 17 -- any citations to back up this assertion?

**AC5.11** It probably doesn't need a citation. We have adjusted the wording to clarify that this statement is based on the underlying measurements used to calculate the variables, not their utility for a given purpose.

**CM5.11** Because they are derived from monthly extremes rather than averaged daily extremes, these variables represent the full range of temperatures experienced in a given location better than the bioclimatic variables.

**RC5.12** Section 4.3 -- Was NDVI not included in predicting climate? If not, it doesn't really seem relevant for this paper.

**AC5.12** Correct, it was not included as a climate predictor. The goal of the project is to generate and assemble good geographic raster data (not only climate) for Hong Kong. Because all of these data products are now in one place, it will be straightforward for researchers and practitioners to access them.

**RC5.13** Figure 5 -- I find it difficult to derive much meaningful information from this figure. I suggest it could be removed if limited for space.

**AC5.13** We are open to this suggestion and leave it to the editor's discretion.

**RC5.14** Figure 6 -- Beautiful figure!

**AC5.14** Thank you kindly!

**RC5.15** The dataset seems in order and easy to use. Aggregated climate normals from HKO should also be included so this study could be reproducible (if legally acceptable).

**AC5.15** We're not sure this will be possible: the permission received from HKO was to distribute climate predictions only, not the underlying data. However the raw data is available on the HKO website for anyone who wants it. For purposes of reproducibility, perhaps we could provide the normals used on a case-by-case basis.

Response to Anonymous Referee #6

Dear Anonymous Referee #6,
Thank you very much for reviewing the manuscript and providing your feedback and concerns. Below we provide point to point responses (AC) to your comments (RC), as well as changes in the manuscript (CM). Page and line numbers refer to those in the the first manuscript revision. We also provide an attached pdf document showing tracked changes.

On behalf of the authors,
Brett Morgan

**RC** - Referee comment        **AC** - Author comment        **CM** - Change in the manuscript

Review of the manuscript "New 30 m resolution Hong Kong climate, vegetation, and topography rasters indicate greater spatial variation than global grids within an urban mosaic" by Morgan and Guénard

General comments

**RC6.01** The manuscript "New 30 m resolution Hong Kong climate, vegetation, and topography rasters indicate greater spatial variation than global grids within an urban mosaic" describes a high- to medium-resolution dataset of a large variety of topography, vegetation and climate rasters for the area of Hong Kong. The authors explain well the motivation and the usefulness of such a dataset emphasizing the applicability especially in Species Distribution Modeling. The selection of different variables, their elaboration and their evaluation are described in detail. While I cannot evaluate if the data manipulation was properly designed and following the standard manipulation procedures, the authors make a great effort to describe their executed procedure in detail. The vast and varied dataset along with the manuscript fit well into the scope of the journal "Earth System Science Data" and could be considered for publication after the authors address some of the comments and technical corrections.

**RC6.02** The dataset DOI link works seamlessly and the reference to the discussion paper is provided on the dataset landing page. The authors could include a short instruction on how to cite the discussion/final paper as well as the dataset itself (consider some entries on the Pangaea repository (https://www.pangaea.de/) for nice examples). The few randomly selected datasets download and open (in two different GIS programs without any problems. The dataset names correspond to the descriptions in the discussion paper. However, on the Figshare page I was not able to locate the monthly zip files and the "readme" document (with file names, descriptions and summary statistics) that the authors describe in the "Data availability" section. The authors should upload these files on Figshare or modify the manuscript.
**AC6.02** We have added recommendations for citing both the dataset and manuscript in Figshare repository. The compressed monthly models are called "all_monthly_models.zip," and are present in the repository when we view it. The summary document is called "_table_of_raster_descriptions.pdf" and is the last file in the repository.

Specific comments

**RC6.03** Title: For me personally the second part of the title ("indicate greater spatial variation than global grids within an urban mosaic") is a bit redundant as it is common knowledge that finer resolution documents variation much better than a coarser resolution. In my opinion, the first part of the title perfectly describes the authors contribution and is adequate on its own. That said, I do not insist on changing the title and just provide my opinion.

**AC6.03** We are not so sure the higher spatial variation (as measured by standard deviation) in our model results are as predictable as you suggest. Would simply aggregating a higher resolution raster into a lower resolution decrease the standard deviation of values? Preliminary analysis of an example (biovars 5) suggests not, because the standard deviation values of our raster before (1.215) and after (1.200) resampling to 1km are similar, and both much higher than that of the corresponding WorldClim raster (0.740). This is also illustrated in Figure 7, where it is clear that differences between our predictions and WorldClim are not due to differences in resolution alone. Perhaps modifications to the title could better communicate that the greater variation found is independent of the raster resolution.

**RC6.04** P1 L2: Maybe "including" would be a more appropriate word than "particularly" in this context.
**AC6.04** We have adopted this suggestion.
**CM6.04** The recent proliferation of high quality global gridded GIS datasets has spurred a renaissance of studies in many fields, including biogeography.

**RC6.05** P4 L8-12: Here you basically reiterate the motivation you already explained in P3 L9-15. I suggest you remove the part on P4 or include some of the text from P4 in P3.
**AC6.05** Yes, it is redundant.
**CM6.05** Removed from P4: "We hypothesize that in addition to providing this finer resolution, our new climate data will indicate greater variation (measured as raster standard deviation) in climate variables than currently available global data products."

**RC6.06** P6 L17-19 and Table 2: The abbreviations of the variables from Table 2 should be specified in the manuscript (for example: "maximum temperature (tmax)") or/and in the table caption.
**CM6.06** Added variable abbreviations in Appendix A: Glossary of variable definitions.

**RC6.07** P6 L22-23: "When necessary, each predictor was statistically transformed to approach a normal distribution" What was the criteria you used to determine, whether it was necessary to transform a predictor?
**AC6.07** This was done somewhat subjectively by viewing histograms of the candidate variables, and transforming them depending on the skew direction. One exception to this was elevation which we did not transform, because elevation and temperature should be linearly related.

**RC6.08** P6 L29: The "AIC" abbreviation is not explained.
**CM6.08** All predictors were initially included, then using the step() function, pared down in each regression model using stepwise bidirectional selection based on the Akaike information criterion, using 4 degrees of freedom as a penalty to make predictor selection stricter than the default.

**RC6.09** P7 L25: Maybe provide a reference for the standard equation?
**AC6.09** We have cited a review that includes the equation as well as discussion of using NDVI in environmental change research.
**CM6.09** NDVI calculations were completed using the standard equation (Pettorelli et al., 2005):
…

**RC6.10** Section 4.2: I have no experience in modeling climate interpolation, so I will not comment on the technical aspects and used variables, which, nevertheless, seem to be sufficiently described in Section 3.2. However, I do have some problems with understanding the climate interpolation modeling results. You describe monthly results of the climate

variables, but I do not understand if this means monthly averages for a period of approx. 20 years (e.g. all the Januaries between 1998 and 2017) or monthly averages for every year (e.g. the average of a variable for January 1998). I think you refer to the first case, but in order to make the manuscript clearer, you should emphasize the considered period in parts of the manuscript and in the figures showing the results.

**AC6.10** You are correct: they represent, for example, all the Januaries from 1998 to 2017.

**CM6.10** As an example, one of these models represents minimum temperatures recorded in all Januaries with data available from 1998 to 2017.

**RC6.11** P8 L15: Why "Minimally"? Do you mean that 32,024 was the lowest used number of measurements for one of the variables? I suggest you rephrase the sentence, to make it clearer.

**AC6.11** This number is an estimate of the total number of weather station measurements used in this study. The uncertainty is due to the threshold of number of years of data required for a weather station to be included in the models: at least 8 years but up to 20. The 32,024 values comes from 12 months x 10 variables x 8 years x the number of stations with data, which varies by variable and month (Table 2). Because 8 years is the lower limit, this number is the minimum total number of measurements used.

**CM6.11** Minimally, a total of 32,024 monthly weather station measurements...

**RC6.12** P8 L20: Why are only 3 of the 10 variables shown in Fig. 5?

**AC6.12** Only 3 variables were included in the interest of saving space. Also if the non-temperature variables were portrayed here, additional color keys would be needed and could risk overcomplicating the figure.

**RC6.13** P9 L29: Should you even refer to Fig. 1 in this part of the manuscript?

**AC6.13** This was an error remaining from before the current Fig. 1 was added.

**CM6.13** (Figs. 2, 7; Table 3)

**RC6.14** Section 4.2.4: When you compare your models with the WorldClim 2, you do not specify if both models cover the same (or at least a similar) time period from 1998 to 2017. Considering the changing climate, it is important to compare climates in the same time frame. If the models cover different time periods, you can still compare the models, but have to discuss the differences between them in light of the different time frame.

**AC6.14** Because our comparisons between the two datasets focus on variation based on standard deviations within each raster, we believe this is less of a concern than if we were considering mean or median raster values, for example. But we do agree it is still a good idea to acknowledge the temporal discrepancy: WorldClim 2 uses data from 1970-2000.

**CM6.14** Though there is a temporal discrepancy between weather station data used in WorldClim 2 (1970-2000) and this study (1998-2017), climate change is unlikely to explain the observed differences in temperature variability. Evidence suggests that if anything, mountains are experiencing climate warming faster than low elevation areas (Pepin et al., 2015), which would give the opposite results of our findings where mountains are cooler than WorldClim indicates (Fig. 7a).

**RC6.15** P11 L28-29: The sentence "Models projecting future climate scenarios would enable biodiversity change predictions, with additional variables like cloud cover and solar radiation useful." is incomprehensible. Probably the comma and "useful" are remnants from a previous version of the manuscript?

**AC6.15** We have fixed these errors and elaborated on how filling data gaps for Hong Kong would benefit research in the region.

**CM6.15** Projections of future climate scenarios could complement historical data to enable predictions of biodiversity change. Additional variables like cloud cover and solar radiation would especially benefit studies of photosynthetic taxa.

**RC6.16** References: As a reader I would prefer to have DOI's (where available) included in the reference list. However, I do not know if DOI inclusion is obligatory in Earth System Science Data.
**CM6.16** Added DOIs to applicable references.

**RC6.17** Table 2: It should be clearly indicated in the table and/or in the table caption, that SD is standard deviation. Additionally, from what was the ratio in this table calculated? From the description in the manuscript I suppose it is the ratio of the standard deviations of the two rasters, however the calculations are off (for example, in the first line: 1/0.5=2 and not 1.9)
**AC6.17** The ratio is indeed the ratio of the SD of the two corresponding rasters, and rounding causes the ratio values appear to be off at times. We have added to the caption to explain this, and clarify that the values are standard deviation. The "Ratio" column has also been renamed to "Increase Ratio."
**CM6.17** Comparisons of variation between bioclimatic variables, measured as raster value standard deviation. All new rasters are more variable than their corresponding Worldclim 2 layers. Increases in standard deviation range from 1.4x to 3.4x. Calculations may appear inaccurate due to rounding.

**RC6.18** Table 2 Caption: The last sentence "Increased standard deviation ranges from 1.4x to 3.4x." is not very clear, as it is not explained which standard deviation the authors have in mind. Additionally, standard deviation values in Table 2 are much larger than 3.4. If the authors refer to the ratio, they should firstly calculate it again (see previous comment).
**AC6.18** See AC6.17.

**RC6.19** Figure S1: I really like this figure as it showcases the whole extent of the work the authors have done. In order to emphasize the several unprecedented datasets that the authors created, they could visually discriminate (by color maybe) between the datasets from other sources and the datasets the authors created themselves (for example: Elevation (30 m) vs. all the Relative Elevations)
**AC6.19** Thank you! Though we agree it would be good to emphasize the highly original data now available, this would include almost all of the boxed items. So colorizing these items likely wouldn't add much utility to the figure.

Technical corrections

**RC6.20** P1 L2: "However, these data, …" instead of "However these data, …"
**AC6.20** We adopted this suggestion.
**CM6.20** However, these data, often 1 km at the finest scale available, are too coarse for applications such as precise designation of conservation priority areas and regional species distribution modeling, or purposes outside of biology such as city planning and precision agriculture.

**RC6.21** P3 L11: "… Kong in …" instead of "… Kong, in …"
**AC6.21** We adopted this suggestion.

**RC6.22** P5 L18: "… the raster package …" should probably be "… the R raster package …"?

**AC6.22** We prefer the format "___ R package" and ensured this is used consistently throughout the manuscript. We also italicized all package names.
**CM6.22** ...the raster R package...

**RC6.23** P5 L19: Probably "Secondly, …" instead of "Second, …"
**AC6.23** Grammatically, using "secondly" vs. "second," etc., seems to be only a matter of preference, and we prefer to omit the -ly.

**RC6.24** P5 L19-20: The sentence "Second, water proximity … surrounding a given pixel" is a bit difficult to read and understand. As they continue in the manuscript by using the term radius, they could maybe write something in the line of "… as the percent of land surface within a radius of a given pixel."
**AC6.24** We adjusted the wording to clarify.
**CM6.24** Second, water proximity (including inland water bodies) was calculated as the percent of the area surrounding a given pixel covered by land.

**RC6.25** P6 L3: Perhaps "… vegetated areas have a value of 0." Instead of "… vegetated areas are 0."
**AC6.25** We adopted this suggestion.
**CM6.25** The resulting 'urbanicity' layers were later used as climate predictors. In these rasters, completely impervious locations have a value of 100, while vegetated areas have a value of 0.

**RC6.26** P6 L8: Perhaps "… linear regressions …" instead of "… linear regression …"
**AC6.26** We refer to linear regression as a general, conceptual statistical approach and therefore prefer to keep the word here singular.

**RC6.27** P6 L32: Probably "Firstly, …" instead of "First, …"
**AC6.27** See AC6.23

**RC6.28** P6 L32: Probably "Secondly, …" instead of "Second, …"
**AC6.28** See AC6.23

**RC6.29** P8 L23: "6° C" instead "6°C"
**AC6.29** While throughout the manuscript we do generally add a space between numbers and units, for temperatures, percentages, and chevrons it is commonly accepted that the space can be omitted and we prefer to keep this format if ESSD does not object.

**RC6.30** P8 L24: "> 900 m, < 18° C" instead of ">900 m, <18°C"; "> 24° C" instead of ">24°C"
**AC6.30** See AC6.29

**RC6.31** P9 L4, L6, L10: "> 2500" instead of ">2500"; "< 1600" instead of "<1600"; "52 %" instead of "52%"
**AC6.31** See AC6.29

**RC6.32** P9 L15, L16, L17: "15.5° C" instead of "15.5°C"; "19° C" instead of "19°C"; "90 %" instead of "90%"; "75 %" instead of "75%"
**AC6.32** See AC6.29

**RC6.33** P9 L31: two times "2° C" instead of "2°C"
**AC6.33** See AC6.29

**RC6.34** P10 L 13: "… exception are the … forests at …" instead of "… exception is the … forests, at …"
**CM6.34** The verdant mangrove forests, at sea level, are an exception.

**RC6.35** P10 L20, L26: Probably "Firstly, …" instead of "First, …"; probably "Secondly, …" instead of "Second, …"; "… array of rasters …" instead of "… array rasters …"
**AC6.35** See AC6.23
**CM6.35** Second, we provide a diverse array of rasters...

**RC6.36** P11 L4: "… presented data …" instead of "… data presented …"; probably "Firstly, …" instead of "First, …"
**AC6.36** We adopted the first suggestion, but see AC6.23 for the second one.
**CM6.36** Here we outline how shortfalls of the presented data may be improved in the future.

**RC6.37** P11 L9, L10: Probably "Secondly, …" instead of "Second, …"; "For example …" instead of "Forexample …"
**AC6.37** For the first suggestion see AC6.23. We adopted the second correction.
**CM6.37** For example...

**RC6.38** P12 L5: "… foreseeable …" instead of "… forseeable …"
**CM6.38** ...foreseeable...

**RC6.39** P12 L10: "Morgan and Guénard …" instead of "Morgan Guénard …"
**CM6.39** (Morgan and Guénard, 2018)

**RC6.40** Fig. 7 caption: "average low temperature" is probably "average temperature"?
**AC6.40** No, this is correct as it stands. Average temperature of coldest month would be a different measurement.